

# Cross-validating precipitation datasets in the Indus River basin

Jean-Philippe Baudouin[1], Michael Herzog[1], and Cameron A. Petrie[2]

[1]Department of Geography, University of Cambridge
[2]Department of Archaeology, University of Cambridge

**Correspondence:** Jean-Philippe Baudouin (jpb88@cam.ac.uk, baudouin.jeanphilippe@gmail.com)

**Abstract.**

Large uncertainty remains about the amount of precipitation falling in the Indus River basin, particularly in the more mountainous northern part. While rain gauge measurements are often considered as a reference they are only punctual and subject to underestimation. Satellite observations and reanalysis output can improve our knowledge but validating their results is often difficult. In this study, we offer a cross-validation of 20 gridded datasets based on rain gauge, satellite and reanalysis, including the most recent and little studied APHRODITE-2, MERRA2, and ERA5. This original approach to cross-validation alternatively uses each dataset as a reference and interprets the result according to their dependency with the reference. Most interestingly, we found that reanalyses represent the daily variability as well as any observational datasets, particularly in winter. Therefore, we suggest that reanalyses offer better estimates than non-corrected rain gauge-based datasets where underestimation is problematic. Specifically, ERA5 has proven to be the most able reanalysis for representing the amounts of precipitation as well as its variability from daily to multi-annual scale. By contrast, satellite observations bring limited improvement at the basin scale. For the rain gauge-based datasets, APHRODITE has the finest representation of the precipitation variability, yet importantly it underestimates the actual amount. GPCC products are the only datasets that include a correction of the measurements but remain likely too small. These findings highlight the need for a systematic characterisation of the underestimation of rain gauge measurements.



# 1 Introduction

Throughout the Holocene, the Indus River and its tributaries have provided much of the water needed by the people living in its basin for various purposes (e.g. food, energy, industry). The diversity of use and the risks associated with scarcity or excess of water under variable and changing climatic and socio-economic conditions highlight the importance of water management in

both Pakistan and north-west India (Archer et al., 2010; Laghari et al., 2012). Moreover, the Indus headwaters are an important locus of water storage with numerous glaciers whose current and future change remains uncertain (Hewitt, 2005; Gardelle et al., 2012). Therefore, a comprehensive evaluation of the basin wide water cycle is needed. Studies that have addressed this issue have stressed the uncertainties inherent in the observed precipitation (Singh et al., 2011; Gardelle et al., 2012; Immerzeel et al., 2015; Wang et al., 2017; Dahri et al., 2018).

Gridded products allow a homogeneous spatialisation of precipitation at a river basin-scale for statistical purposes (Palazzi et al., 2013). They can be derived from rain gauges, satellite imagery or atmospheric models (e.g. reanalysis), but need validation to assess their quality. Most studies that validate precipitation products in Pakistan, India, or in the adjacent mountainous areas (Hindu-Kush / Karakoram / Himalayas) make use rain gauge data as a reference, either directly from the weather stations (Ali et al., 2012; Khan et al., 2014; Ghulami et al., 2017; Hussain et al., 2017; Iqbal and Athar, 2018), or after gridding

(Palazzi et al., 2013; Rajbhandari et al., 2015; Rana et al., 2015, 2017). However, some authors have pointed out that these reference datasets also suffer limitations that could dramatically reduce correlation and increase biases, incorrectly lowering the confidence in the dataset validated (Tozer et al., 2012; Ménégoz et al., 2013; Rana et al., 2015, 2017).

The first issue of validating gridded precipitation products with rain gauge measurements is simply the uncertainty of the measurements. Beside the risk of corruption or missing values in the reporting process, it has been demonstrated that rain

gauges can underestimate precipitation (Sevruk, 1984; Goodison et al., 1989). The main source of underestimation is wind-driven under-catchment that can reach up to 50% in case of snowfall (Goodison et al., 1989; Adam and Lettenmaier, 2003; Wolff et al., 2015; Dahri et al., 2018), but also includes wetting of the instrument, evaporation before measuring, and splashing out (WMO 2008). Dahri et al. (2018) used the guidelines from the World Meteorological Organization (WMO) to re-evaluate the precipitation measured from hundreds of rain gauges in the upper Indus and found the underestimation to be between

1 and 65% for each station, and 21% basin wide. The second issue is the one of spatial representativeness. A rain gauge records a measurement at a specific location whereas in a gridded dataset, each value represents the mean over all the grid box. The two types of data thus have a different spatial representativeness. This discrepancy in representativeness increases when considering shorter timesteps and areas with strong heterogeneity such as mountainous terrains, which is especially impactful when studying extreme events. Some methods exist to quantify and tackle this issue (e.g. Tustison et al., 2001; Habib et al.,

2004; Wang and Wolff, 2010).

The gridding method is used to spatially homogenise point measurements and also has limitations. Firstly, the specificity of the interpolation method can impact the result (Ensor and Robeson, 2008; Newlands et al., 2011). Secondly, the sparsity of the weather stations increases the uncertainties, which can range from 15 to 100% in areas with a low number of rain gauges (Rudolf and Rubel, 2005). This last point is especially problematic in the Indus River basin. For climatological purposes, the





WMO has published guidelines for the density of rain gauges: from one station per 900km2 in flat coastal areas, to one every 250 km2 in mountains (WMO, 2008). However, the Meteorological Department of Pakistan have recently published a 50-year climatology of precipitation for the country based on 56 stations, which uses around one station per 15,000 km2 (Faisal and Gaffar, 2012). Gridded rain gauge-based datasets rely on a similar density of observations in the Indus River basin (cf.

Figure 1, Table 4). The situation in India is better as the Indian Meteorological Department produces a country-wide dataset of precipitation that is used for monsoon monitoring and includes up to 6300 stations. This distribution makes around one station per 500km2, which is well within the WMO guideline. However, areas of lower density remain, especially in the western Himalayas and the Thar Desert, which are both in the Indus River basin (Kishore et al., 2016). Rain gauges are not only scarce in mountainous areas, but their location is also biased. In order to be accessible all year long, they are generally situated at

the bottom of valleys, and these locations appear to be significantly drier than locations at altitude (Archer and Fowler, 2004; Ménégoz et al., 2013; Immerzeel et al., 2015; Dahri et al., 2018), which means that the interpolation method underestimates precipitation in the surrounding mountains.

There are a number of ways of overcoming the limitations of gridded rain gauge data, including the use of data derived from satellites and reanalysis. Satellite imagery can help to reduce both the lack and heterogeneity of surface measurements.

Satellite-based products generally make use of global infrared observations of cloud cover and microwave measurements along a swath (the narrow band where the observations are made as the satellite passes). However, their abilities over a heterogeneous terrain are more limited than over a flat and homogeneous areas (Khan et al., 2014; Hussain et al., 2017; Iqbal and Athar, 2018). Moreover, these products still need rain gauges for calibration and are therefore dependent on the quality of those.

Reanalyses of the atmosphere offer another way to estimate precipitation. There are many valuable variables in a reanalysis,

which are the result of the assimilation of observations with model outputs, but their estimates of precipitation are, in most case, a pure model product. That is, the precipitation is a forecast generated by the model used for the reanalysis, and is not constrained by direct observations in the way that other assimilated quantities are. Models are known to predict precipitation with difficulty and most studies consider that precipitation from reanalysis is less reliable than that based on observation (Rana et al., 2015; Kishore et al., 2016). The reasons often invoked include discrepancies in spatial patterns and important biases.

However, recent progress in assimilation techniques has it made possible to integrate precipitation observations in the most recent reanalysis (ERA5, Hersbach et al., 2018), and significant improvements are possible (e.g. Beck et al., 2019).

This study aims to better understand the quality and limitations of 20 precipitation datasets that are available for a domain of study encompassing the Indus River basin. Specifically, it focuses on the underestimation of rain gauge-based products, the possible improvement that can be obtained from remote sensing, and the capability of reanalysis products to represent the

basin wide precipitation variability. Previous studies have investigated the qualities of precipitation datasets in this area (e.g. Ali et al., 2012; Palazzi et al., 2013; Khan et al., 2014; Hussain et al., 2017), but none have looked at such a large number of datasets nor at the most recent ones. Moreover, our method slightly differs, as we offer a cross-validation, thereby avoiding the problems that come from the selection of a unique reference. We cross-compare each of the datasets, identify their similarities and discrepancies, and using the diversity of data source and method, assess their strengths and weaknesses. After presenting



the datasets selected for the study, we give a general description of the methods. The subsequent result section is split into three parts, which review: i) the seasonal cycle and annual means, ii) daily variability, and iii) monthly and longer term variability.





## 2  Data and Methods

### 2.1  Domain of study

The Indus River basin extends across the north-westernmost part of the South Asian sub-continent. It is bounded from the north-east to the west by high mountain ranges, including the Himalayas, Karakoram, Hindu Kush and Suleiman Ranges. To

the south, the Indus river flows into the Arabian Sea. The eastern border is the most ambiguous as it extends into the flat dune-fields of the Thar desert. Much of the precipitation that falls in this extensive area evaporates before reaching the Indus River or the sea. Once on the ground, it forms seasonal rivers, such as the Luni River, which has been included in the domain of study (outer dark blue contour in figure 1-A). This particular river flows into the Rann of Kutch, which is a flat salt marsh with complex connections with the Arabian Sea and the mouth of the Indus River (Syvitski et al., 2013), and is bound on the

west by the Aravalli Range. Although not strictly a part of the Indus watershed, it provides a clear and steady boundary for the domain of study.

Differences in relief and precipitation seasonality and pattern suggest that the basin can be separated into two distinct domains. In the flat southern part, most of the precipitation occurs in July and August, under the influence of the South-Asian summer monsoon propagating from the Indian Ocean and India, while the rest of the year it remains dry (e.g. Ali et al., 2012;

Khan et al., 2014; Rana et al., 2015). By contrast, the northern domain is much more mountainous and encompasses a steep maximum precipitation along the Himalayan front (Figure 1-A). This precipitation falls throughout the year, but exhibits a seasonal bi-modality explained by a differences in process (e.g. Archer and Fowler, 2004; Singh et al., 2011; Palazzi et al., 2013). As it does in the southern part of the basin, a sharp peak in precipitation occurs in July-August related to the summer monsoon, but a second, broader peak also occurs in winter, from January to April, triggered by mid-latitude, extra-tropical

western disturbances (Cannon et al., 2015; Dimri and Chevuturi, 2016; Hunt et al., 2018). We have divided the basin along a line between 68.75°E-33.5°N and 77.5°E-30°N (inner dark blue contour in figure 1-A), which broadly corresponds to the 100mm isohyet of winter precipitation (defined from December to March). The northern part of the basin (hereafter the upper Indus) thus includes the maxima of precipitation along the Himalayas and most of the winter precipitation, while the southern part (hereafter the lower Indus) focuses only on the summer precipitation (Figure 2).

### 2.2  Data

#### 2.2.1  Rain gauge

We have selected five commonly used and one newly available gridded dataset based only on rain gauge data (first six datasets in Table 1). The Asian Precipitation Highly Resolved Observed Data Integration Towards Evaluation of water resources (APHRODITE; Yatagai et al., 2012)) was developed by the Research Institute for Humanity and Nature (RIHN) and the

Meteorological Research Institute of Japan Meteorological Agency (MRI/JMA). Specific to Asia, it is one of the best datasets available for the area (Rana et al., 2015), both in term of resolution (0.25° and daily, it includes a large number of rain gauges; Table 2) and the fact that it covers an extended period (over 50 years). However, it does not provide data after 2007. A new





dataset has been issued in 2019 from the same institute with a period covered extending to 2015 and using a new algorithm (APHRODITE-2), though its quality has not yet been investigated . Covering the whole twentieth century at a monthly resolution, the Global Precipitation Climatology Center monthly dataset (GPCC-monthly; Schneider et al., 2018) is widely used in climatology and for calibration purposes (e.g. satellite-based datasets, Table 1). GPCC-daily (Ziese et al., 2018) offers a

better temporal resolution (daily), but at a lower spatial resolution and has a much-reduced time coverage compared to GPCC-monthly. It uses a smaller number of rain gauges (Table 2), but is constrained by GPCC-monthly. The precipitation dataset from the Climate Research Unit (CRU; Harris and Jones, 2017), has a similar resolution and time coverage to GPCC-monthly, which is useful for comparison. We also selected another daily dataset from NOAA's Climate Prediction Center (CPC; Xie et al., 2010). Although CPC uses a lower number of rain gauges compared to APHRODITE (Table 2), its availability extends

to the present with near real time updates, which means that it can be used for calibrating other near real time products (e.g. CMAP in Table 1 and MERRA2 in Table 3).

### 2.2.2   Satellite

We selected four of the available satellite-based gridded precipitation products (last four datasets in Table 1). The Tropical Rainfall Measuring Mission (TRMM) Multi-satellite Precipitation Analysis (TMPA; Huffman et al., 2007) has the highest

temporal and spatial resolution of the available satellite datasets (sub-daily, and 0.25° like APHRODITE and GPCC-monthly) and the largest variety of input. The daily product from the Global Precipitation Climatology Project (GPCP-1DD; Huffman and Bolvin, 2013) is useful for comparison. The same project also issued a monthly dataset (GPGP-SG Adler et al., 2016). All three of these datasets use GPCC for calibration. We have also included CPC Merged Analysis of Precipitation (CMAP; Xie and Arkin, 1997), which uses CPC for calibration, and has the same time coverage and resolution as GPCP-SG. We have

selected the version that does not include reanalysis data, to simplify the analysis.



**Table 1.** Observational datasets of precipitation selected for this study, derived from rain gauges or satellites

| Name | Version | Time coverage | Time resolution | Spatial resolution | Based on | Reference |
|------|---------|---------------|-----------------|--------------------|----------|-----------|
| APHRODITE | V1101 | 1951-2007 | Daily | 0.25° | Rain gauge only | Yatagai et al. (2012) |
| APHRODITE-2 | V1901 | 1998-2015 | Daily | 0.25° | Rain gauge only | |
| CPC | V1.0 | 1979 (monthly) / 1998 (daily) -2018 | Daily | 0.5° | Rain gauge only | Xie et al. (2010) |
| GPCC-daily | V2 | 1982-2016 | Daily | 1° | Rain gauge and GPCC-monthly | Ziese et al. (2018) |
| GPCC-monthly | V8 | 1891-2016 | Monthly | 0.25° | Rain gauge only | Schneider et al. (2018) |
| CRU | TS4.02 | 1901-2017 | Monthly | 0.5° | Rain gauge only | Harris and Jones (2017) |
| TMPA | 3B42 V7 | 1998-2016 | 3-hourly | 0.25° | GPCC, satellites | Huffman et al. (2007) |
| GPCP-1DD | V1.2 | 1996-2015 | Daily | 1° | GPCC, satellites | Huffman and Bolvin (2013) |
| GPCP-SG | V2.3 | 1979-2018 | Monthly | 2.5° | GPCC, satellites | Adler et al. (2016) |
| CMAP | V1810 | 1979-2018 | Monthly | 2.5° | CPC, satellites | Xie and Arkin (1997) |





**Table 2.** Number of stations used on average for the rain-gauge-based datasets (except CRU for which this information was not directly available), per time step, for the two study areas, and over the period 1998-2007.

| Datasets | Upper Indus | Lower Indus |
|---|---|---|
| APHRODITE | 55 | 48 |
| APHRODITE-2 | 88 | 65 |
| CPC | 15 | 21 |
| GPCC-daily | 11 | 16 |
| GPCC-monthly | 35 | 33 |



### 2.2.3 Reanalysis

Unlike the observation datasets, reanalysis datasets can be quite different from one another. They generally use their own atmospheric model and assimilation scheme, and the type and number of observations assimilated varies. Therefore, we selected a large set of ten reanalysis datasets (Table 3). The four reanalyses of the latest generation are, from most recent to oldest:

ERA5 (Hersbach et al., 2018) from the European Centre for Medium-Range Weather Forecasts (ECMWF), the Modern Era Retrospective-analysis for Research and Applications version 2 (MERRA2; Gelaro et al., 2017) from the NASA, the Japanese 55-year Reanalysis (JRA; Kobayashi et al., 2015) from the JMA, and the Climate Forecast System Reanalysis (CFSR; Saha et al., 2010, 2014) from the National Center for Environmental Prediction (NCEP). These are still regularly updated, and they all include the latest observations from satellites and cover the full satellite era from at least 1980. JRA goes back to 1958,

when the global radiosonde observing system was established, while ERA5 will eventually cover the whole second half of the twentieth century.

In terms of technical differences, ERA5 uses a more complex assimilation scheme than the others (4DVAR), which allows for better integration of the observations. It is also the only one that assimilates precipitation measurements. MERRA2 also uses observations, but takes them from a gridded dataset (CPC) and only uses them to correct the precipitation field before

analysing the atmospheric impact on the land surface; this changes land surface feedbacks on the atmosphere. CFSR is an Ocean-Atmosphere coupled reanalysis, that is, the sea surface is modelled and provides feedback to the atmospheric model, instead of being prescribed by an analysis from observations. ERA5 and MERRA2 are the most recent of the reanalysis datasets to be published, and not many studies have looked at the improvement from their predecessor, respectively ERA-Interim (Dee et al., 2011) and MERRA1 (Rienecker et al., 2011). Both have stopped being updated or will be very shortly, but they are

included in this study for comparison purposes.

Reanalyses for the whole twentieth century have also been produced, but to retain the homogeneity of the type of observations assimilated they only include surface observations. The twentieth century reanalysis from NCEP (20CR; Compo et al., 2011), only assimilates surface pressure, but more recently, the ECMWF produced ERA-20C (Poli et al., 2016), which has surface wind assimilated along with surface pressure.

We have also made use of older generation reanalysis datasets that are still being updated, including: the NCEP/NCAR reanalysis (NCEP1; Kalnay et al., 1996) and the NCEP/NDOE reanalysis (NCEP2; Kanamitsu et al., 2002). Both are useful to quantify the progress in reanalysis systems as well as to compare them with more observation-limited century long reanalyses.



**Table 3.** Datasets of precipitation selected for this study, derived from reanalysis

| Name | Time coverage | Spatial resolution | Remarks | Reference |
|---|---|---|---|---|
| ERA5 | 1979-2018 | 0.25° | 4DVAR, precipitation assimilated | Hersbach et al. (2018) |
| ERA-Interim | 1979-2018 | 0.75° | 4DVAR assimilation scheme | Dee et al. (2011) |
| JRA | 1958-2018 | 0.5° | | Kobayashi et al. (2015) |
| MERRA2 | 1980-2018 | 0.5° / 0.625° | Correction of the precipitation with CPC for land interaction. Assimilate aerosol observations | Gelaro et al. (2017) |
| MERRA1 | 1979-2010 | 0.5° / 0.66° | | Rienecker et al. (2011) |
| CFSR | 1979-2018 | 0.5° | Coupled reanalysis (atmosphere, ocean, land, cryosphere). Same analyses as MERRA1. Version 2 starting in 04/2011 | Saha et al. (2010, 2014) |
| NCEP2 | 1979-2018 | 1.875° | Fixed errors and updated model since NCEP1 No satellite radiance assimilated | Kanamitsu et al. (2002) |
| NCEP1 | 1948-2018 | 1.875° | No satellite radiance assimilated | Kalnay et al. (1996) |
| 20CR | 1871-2012 | 1.875° | Assimilate surface pressure only | Compo et al. (2011) |
| ERA-20C | 1900-2010 | 1° | Assimilate surface pressure and marine wind only | Poli et al. (2016) |



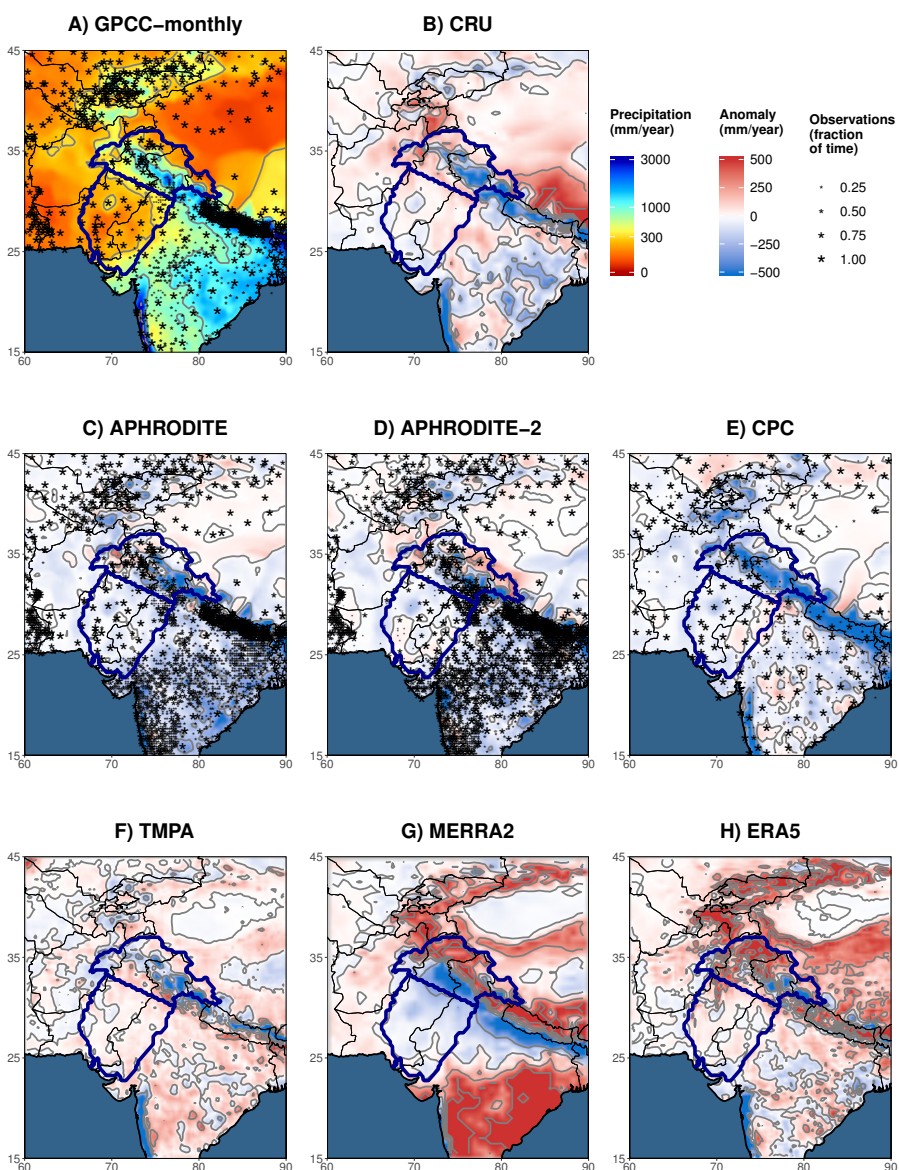

**Figure 1.** Map of annual mean precipitation for different datasets. The annual mean is computed over the period 1998-2007. GPCC monthly (A) is used as a reference to compute the anomaly for the other datasets (B to H). The grey lines are the isohyets whose level corresponds to the labels in the legend. The boundaries of the two study areas are displayed in dark blue on each map. The stars mark the grid cells that include at least one gauge observation. The size of the stars represents the number of time steps with at least one observation over that cell, relative to the total number of time steps needed to compute the annual mean (120 for A, 3652 for C,D and E). This information was not available for CRU (B), and does not apply to the satellite-based TMPA (F), and the two reanalyses (G and H).





## 2.3 Methods

For each dataset, the time series of precipitation averaged over the two domains of study (upper and lower Indus) were calculated at a monthly resolution, and daily if possible. In order to keep the boundaries of the domains of study fixed, all data were bi-linearly interpolated to a 0.25° grid, common to APHRODITE, APHRODITE-2, and GPCC-monthly.

Most of the analysis is performed over the 10-year period from 1998-2007, which is common to all of the datasets. Different timescales are investigated: daily, monthly, inter-annual and decadal, as well as the seasonality. Summer is defined from June to September for both domains, which matches the monsoon precipitation peak. Winter is defined from December to March for the upper Indus, which fits the snowfall peak rather than the precipitation peak, but this makes it possible to focus on the issues on snowfall estimation (Palazzi et al., 2013). Winter is not defined in the lower Indus, as it is a dry season.

The study mainly focuses on comparing the mean and variability of the time series. For the variability, the Pearson coefficient is mainly used, along with the Spearman coefficient in case of issues with extreme values. The differences between the time series are explained by the differences in the way the datasets are produced: either from the method or from the raw data used in input. This helps with identifying and quantifying the key sources of uncertainty. Conversely, if two datasets share similarities, it is often due to similarities in the methods or in the input data. Taking those datasets into account to evaluate the uncertainty

will lead to its underestimation. However, similarities are also shared by unrelated datasets. When two time series with no common input data and different methods exhibit a common behaviour, it is likely due to the fact that behaviour is present in the real signal. Therefore, we assume that the two time series showing common behaviour better represent the reality than a more dissimilar third one, which eventually helps reducing the uncertainty.

    The use of observational datasets and reanalyses in combination is key in this study. Reanalyses use very different methods

in their production than the gridded observational datasets, and most do not include precipitation observations in their input. Reanalyses and observational datasets are thereby independent and can be used to validate one another. This is made possible by the increase in quality of the most recent reanalyses, as we will now discuss in the result section.



## 3   Results

### 3.1   Seasonal cycles and annual means

The study of the seasonal cycle has been performed through the analysis of the monthly (Figure 2), seasonal, and annual mean (Table 4) of precipitation over the two domains of study. The results are split between observations (based on both rain gauges

and satellites) and reanalyses. To simplify the analysis, we compare the datasets to GPCC-monthly data, as it likely provides good estimates (see below).

All the datasets record the wet and dry seasons, although with different magnitudes. For the upper Indus, the three rain gauge-based datasets: CPC, APHRODITE and GPCC-monthly, are ranked in the same order for each month (Figure 2-A). CPC is the driest (-51% in annual mean compared to GPCC-monthly), followed by APHRODITE (-23%). Similarly, CPC

and APHRODITE are drier than GPCC-monthly in the lower Indus, although in a different order (-15% and -22% respectively, Figure 2-C). By contrast, GPCC-daily and CRU are much closer to GPCC-monthly (difference inferior to 5 and 8% respectively).

Several reasons explain the differences of mean precipitation between these datasets. First, GPCC products include a correction of the rain gauge measurements of about +5 to +10% (cf. Figure 4 in Schneider et al., 2014). Second, the datasets have

different resolutions that affect the spatial representativeness, despite having carefully interpolated all datasets to the same grid. For instance, GPCC-daily is based on GPCC-monthly, therefore their differences in climatology should mainly be due to the difference in spatial resolution. Third, the interpolation method affects the spatial patterns and can lead to basin-scale differences. APHRODITE includes a method that considers the orientation of the slope to quantify the influence of nearby stations. This greatly reduces the amount of precipitation falling in the inner mountains compared to GPCC-monthly. An example of

this pattern is evident for the northern side of the Himalayas where only very few observations exist (Figure 1-D; Yatagai et al., 2012). In CRU, the interpolation method (triangulated linear interpolation of anomalies; Harris et al., 2014) seems to smooth areas of strong gradients such as near the foothills of the Himalayas (Figure 1-B). This smoothing would explain a slightly drier upper Indus, and slightly wetter lower Indus, compared to GPCC-monthly (Table 4). Fourth, the number of stations used to compute the statistics varies dramatically from one dataset to another, particularly in the upper Indus where the precipitation

patterns are the most heterogeneous (Figure 1, Table 2). CPC is likely drier because of a lower number of observations, leaving vast areas with no or very few observations, including the wettest regions (Figure 2-E). However, there is no linear relation between precipitation quantity and number of observations. GPCC-daily includes the lowest number of observations and still has an annual mean precipitation over the domain of study that is similar to GPCC-monthly, which is due to its being constrained by GPCC-monthly. On the contrary, APHRODITE has a much higher number of observations, but remains much drier than

GPCC-monthly. APHRODITE's creator pointed out that the differences in quality checks could explain this behaviour (Yatagai et al., 2012). This possibility is especially so as the product partially relies on GTS data that are sent in near real time to the global network, which means that there is a significant risk that missing values are reported as no precipitation, thus leading to underestimation of the total precipitation. In that case, the issue should also affect CPC, which is based on GTS data only. In GPCC-monthly (and daily), only stations with at least 70% of data per month are retained (Schneider et al 2014), while in





CRU this number is increased to 75% (Harris et al., 2014). Thus, limiting the analysis to the most reliable weather stations could help build confidence in the total precipitation amount. However, we were not able to prove the existence of spurious null or close to zero values at grid points with observations that are compared to TMPA, due to too low correlation. Rather, we found strong differences on the monthly value of GPCC, APHRODITE and CPC, at grid points where the three have integrated observations. This could suggest that the three datasets use differently consolidated precipitation measurements from the same weather station, or handle the interpolation at a grid point with observations differently.

Interestingly, APHRODITE-2 has a higher mean than APHRODITE (+15% and +13% in the upper and lower Indus respectively), which is closer to GPCC-monthly. Different changes have been performed on the methodology: quality control of extreme high, station-value conservation, merging daily observation with different definitions for the start and end of a day, and updated climatology. However, the difference in mean precipitation is mainly related to the change in observations from rain gauges: they are basin-wide more numerous, but this increase mainly happens over Indian territory. Pakistan actually sees a decrease in precipitation measurements, especially in the dry southern central part (Figure 1-D). This decrease in observations in the drier area explains the increase in mean precipitation in the lower Indus. In the upper Indus, the increase is mainly due to the inclusion of measurements from one isolated weather station in the northernmost part of India.

The mean summer precipitation of the satellite-based datasets overall differs little from GPCC-monthly estimate for the upper Indus (less than 3%, Table 4). In winter, TMPA, GPCP-1DD, and GPCP-SG have slightly lower estimates than GPCC-monthly, and are more comparable to APHRODITE. CMAP stands out, reporting the largest winter precipitation amount (+36% compared to GPCC-monthly). In the lower Indus, all satellite products are wetter than the rain gauge-based products (from 10 to 30% more than in GPCC-monthly in the annual mean), especially for both GPCP products. The complexity of the algorithm used to produce the satellite-based datasets makes the reasons for their differences with each other or with rain gauge products unclear. According to previous studies, their ability to represent precipitation over rough terrain such as the upper Indus is limited (e.g. Hussain et al., 2017). In contrast, precipitation estimates over flat terrain with sparse observations and mostly convective precipitation like the lower Indus benefit from satellite observations (Ebert et al., 2007).

The reanalyses represent the dry and wet seasons, but with a somewhat larger spread than the observations (Figure 2-B and D). This is especially true for the lower Indus, with two outliers: JRA, which is wetter by a factor of two; and 20CR, which has almost no wet season. The monthly variations are, on the other hand, generally well captured, with a maximum in July. For the upper Indus, some discrepancies are evident in the seasonality. The most striking one is a wetter winter. On average winter precipitation is 30% higher than in GPCC-monthly, with the notable exception of ERA-20C (Table 4). Those wetter conditions also extend to the surrounding drier months: April-May and October-November. Interestingly, the mean summer precipitation for the reanalyses is not significantly wetter than GPCC-monthly (Table 4). Only Era-Interim stands out with a wet bias, mainly in the north-west corner of the upper Indus domain, a bias partly corrected in ERA5 (Figure 1-H). Overall, the annual precipitation for the upper Indus is approximately 20% higher in the reanalysis products than in GPCC-monthly.

A second discrepancy is visible between a majority of the reanalyses and the observations for the upper Indus: a delay in the seasonality starting during the pre-monsoon season. The observations show that May is the driest month of that season and is followed by a sharp increase in precipitation in June, but only ERA5, ERA-Interim, and MERRA1 represent this behaviour. In





contrast, NCEP2 and CFSR are much drier in June than in May. For the other reanalyses, the precipitation for May and June are comparable. This delay continues into the summer monsoon period: while the observations clearly show a wetter July than August, this is only the case for ERA5, ERA-Interim, and both MERRA reanalyses. A similar delay can be found over the Ganges plain and all along the Himalayas, which suggests wider uncertainties on the monsoon propagation in the reanalysis.

Difference in winter precipitation between reanalysis and observational datasets can at least in part be explained by a dry bias in the observations. Recently, Dahri et al. (2018, hereafter Dahri2018) compiled the measurements from over 270 rain gauges in the upper Indus and adjusted their results to undercatchment, following WMO guidelines. They found a basin-wide adjustment of 21%, but this varies from 65% for high altitude stations, to around 1% for the ones in the plain. GPCC-monthly, the only observational dataset to take undercatchment into consideration, adjusts the values by around 5 to 10% (Schneider et al., 2014).

Therefore, the Dahri2018 study indicates that GPCC correction factors are largely underestimated in the mountains. Dahri2018 selected a domain of study somewhat smaller than the upper Indus domain presented here, and covers the period 1999-2011. To make comparison with its results we recomputed the annual mean of several of the most recent and highest resolution datasets to fit these requirements (Table 5). We found that the unadjusted precipitation mean in Dahri2018 is 7% lower than the GPCC value, a percentage around the correction factor used in GPCC. This suggests that the unadjusted values in the two datasets

are very close, while GPCC-monthly uses much fewer stations measurements. While this could highlight GPCC quality, we also found some discrepancies in the spatial patterns that offset each other at the basin scale. In the Karakoram Range first, at the northernmost part of the upper Indus domain, GPCC-monthly exhibits lower precipitation means than in Dahri2018, which cannot be explained by the difference of correction factor between the two datasets. The nearest stations used in GPCC-monthly are all located in the dry and more accessible Indus River valley to the south of the mountain range (Figure 1-A). Those drier

conditions are extended to the north by the interpolation method, while Dahri2018 integrate station measurements suggesting wetter conditions than in the valley. This approach illustrates the impact of biased weather station locations mentioned in the introduction and in several previous studies ( e.g. Archer and Fowler, 2004; Ménégoz et al., 2013; Immerzeel et al., 2015). In contrast, in the western part of the domain, GPCC-monthly extends the precipitation maximum to the north, which leads to an overestimation compared to the precipitation measurements used in Dahri2018. Eventually, those two biases seem to offset

each other, making the area averaged estimate of GPCC-monthly close to Dahri2018. APHRODITE-2 and TMPA similarly fail at representing the differences between valleys and mountains and underestimate significantly the adjusted value from Dahri2018 (-21% and -31% respectively).

      By contrast, the four selected reanalysis datasets in Table 5 overestimate the Dahri2018 adjusted value. We noted that the south border of the domain of study in Dahri2018 overlaps with the highest precipitation rates, which occur in summer.

Depending on the different representation of the relief and its modelled impact on weather in the reanalyses, a small shift to the north or the south can occur, with a potentially important impact on the mean over Dahri2018's domain. In contrast, the upper Indus domain we have delineated includes all the highest rates of precipitation falling in the Indus River basin, which reduces the impact of a possible small shift to the north or the south of the maxima. A shift to the north is especially evident in MERRA2 (Figure 2-G) and explains the stronger overestimate found in Table 5 than in Table 4, compared to GPCC-monthly.





All rain gauge-based datasets thus suffer from underestimating annual mean precipitation for the upper Indus when compared to Darhi2018. The main reason is linked to the adjustment of undercatchment, which is either underestimated or is not sufficiently considered. This is especially critical in winter. Differences in quality control and interpolation method also likely explain differences between the rain gauge-based datasets and impact both parts of the basin. Those errors are more consistent

yearlong. Interestingly, interpolation uncertainty seems not limited to areas with low density observations. Further, satellite observations might bring improvement in areas with sparse observations. For example, a higher precipitation mean for the lower Indus is possibly due to better consideration of local higher rates during convective events. However, satellite products cannot correct observation biases since they use them for calibration, and biases remain unchanged for the upper Indus domain. Overall, GPCC-monthly has the closest estimate to Dahri2018, thanks to a large ensemble of good quality rain gauge measure-

ments. A study with a high density of bias-corrected rain gauge measurements would be needed in the lower Indus to identify which dataset better represents the precipitation mean. The IMD (Indian Meteorological Department) dataset can provide such reference for the Indian part of the basin. Kishore et al. (2016) showed that the IMD product is closer to GPCC-monthly than to APHRODITE in the north-west part of India, while in Prakash et al. (2015) TMPA biases are limited between -10 and 10% in that domain.

Except for ERA5 and MERRA2, reanalysis data are independent from rain gauge measurements and are therefore not affected by these underestimates. They suggest a much wetter winter season for the upper Indus, while the latest reanalyses (ERA5, JRA, MERRA2, and CFSR) converge towards a similar amount, which is possibly closer to the reality than observation-based estimates, although some over estimations cannot be overruled. Summer is a more complicated season, the spread of precipitation intensity in reanalysis products is high to very high, varying between -29% and +48% of GPCC-monthly mean

for the upper Indus, and between -72% and +164% for the lower Indus. Even when only considering the latest reanalyses, the summer mean does not converge. Moreover, their seasonality does not always fit the observations, nor their representation of spatial patterns. These latest difficulties are detrimental to small-scale comparisons, especially near mountains, but justify our choice to use a larger study domain. MERRA2 and ERA5/ERA-Interim have estimates of the pre-monsoon and monsoon seasonality closer to observations in the upper and the lower Indus, respectively.





**Table 4.** Mean annual and seasonal precipitation (in mm) falling over the two study areas, for the period 1998-2007. Winter is defined from December to March and summer from June to September. The first ten datasets are observations, the second ten are reanalyses.

| Datasets | Upper Indus | | | Lower Indus |
| --- | --- | --- | --- | --- |
| | Winter | Summer | Annual | Annual |
| APHRODITE | 154 | 237 | 484 | 198 |
| APHRODITE-2 | 179 | 272 | 555 | 223 |
| CPC | 98 | 200 | 355 | 216 |
| GPCC-daily | 201 | 297 | 607 | 243 |
| GPCC-monthly | 201 | 301 | 613 | 255 |
| CRU | 166 | 281 | 565 | 267 |
| TMPA | 156 | 298 | 555 | 286 |
| GPCP-1DD | 161 | 305 | 569 | 317 |
| GPCP-SG | 167 | 309 | 583 | 325 |
| CMAP | 273 | 307 | 696 | 279 |
| ERA5 | 280 | 380 | 828 | 300 |
| ERA-Interim | 289 | 445 | 931 | 305 |
| JRA | 299 | 325 | 810 | 586 |
| MERRA2 | 265 | 310 | 724 | 177 |
| MERRA1 | 205 | 267 | 598 | 355 |
| CFSR | 282 | 214 | 656 | 162 |
| NCEP2 | 274 | 259 | 703 | 276 |
| NCEP1 | 372 | 343 | 915 | 239 |
| 20CR | 244 | 319 | 746 | 116 |
| ERA-20C | 175 | 276 | 551 | 175 |



**Table 5.** Mean annual precipitation (in mm) for various datasets over the study area defined in Dahri et al. (2018) for the period 1999-2011. Both adjusted and unadjusted values (the latter in parenthesis) from Dahri et al. (2018) are reported in the second line

| Datasets | Upper Indus |
|---|---|
| Dahri2018 | 697 (574) |
| APHRODITE-2 | 548 |
| GPCC-monthly | 612 |
| TRMM | 480 |
| ERA5 | 835 |
| JRA | 827 |
| MERRA2 | 929 |
| CFSR | 783 |



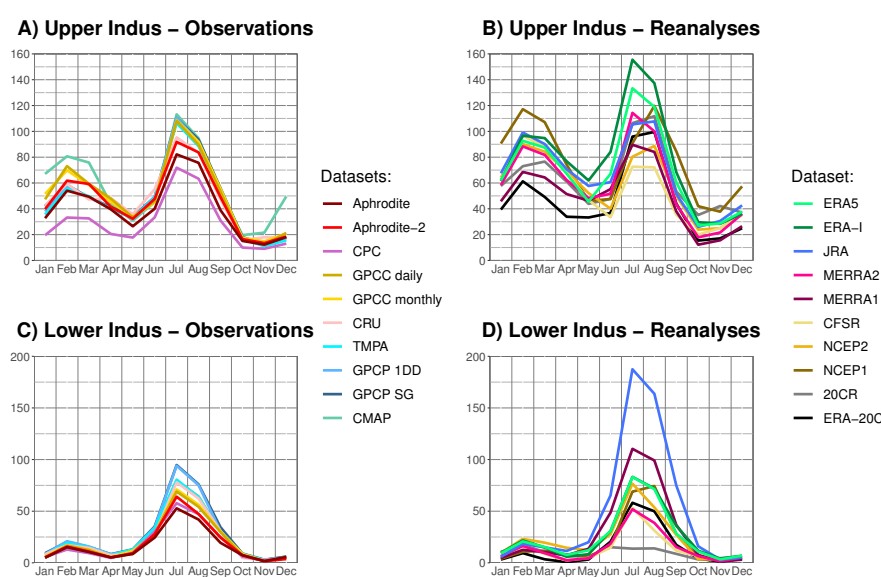

**Figure 2.** Monthly mean of precipitation, over the period 1998-2007, representing the seasonal cycle. Results are split between upper Indus (A and B) and lower Indus (C and D) as well as observation datasets (A and C) and reanalysis (B and D).





## 3.2 Daily variability

Comparing the daily variability helps to quantify the dependency between each dataset, either coming from the use of common methods or input data, or from the representation of the true variability. Before computing the daily correlation, we checked for possible lags between the datasets, using especially the sub-daily resolution of TMPA and most of the reanalyses (e.g. is the

daily value of APHRODITE more correlated with the daily value of TMPA accumulated from 21h, 00h, 03h, etc.). Lags can have different origins. The first is the accumulation period considered for the rain gauge measurements. CPC documentation (Xie et al. 2010) points out that the official period is different from one country to another (in our case, Afghanistan, Pakistan, and India all have different periods, starting at 00hUTC, 06hUTC, 03hUTC, respectively, cf. End of Day time for CPC), which could impact precipitation estimates. GPCC-daily documentation does not mention this issue, while from APHRODITE to

APHRODITE-2, a specific effort has been made to homogenise all observations. Secondly, the TMPA algorithm uses the 00h imagery for the following day accumulation, and therefore, is more representative of an accumulation starting at 22:30h UTC. Thirdly, biases in the daily cycle are possible in the reanalyses.

Our main finding relates to CPC. Figure 3 shows the daily correlation year per year of CPC against APHRODITE and MERRA2, for two lags: 0h and -24h (previous day for CPC). We found that the two lags switch their behaviour somewhere

around 1997/1998, which we interpret as an error in the data processing for CPC. That is, in CPC before 1998, precipitation values correspond to those for the following day. This should not have an important impact on monthly and longer accumulations, but we limited the daily analysis of CPC to the period from 1998 to 2018. Moreover, similar errors might have happened earlier during the 1980s as the curves in Figure 3 come closer or invert again. This error also propagates to the corrected precipitation of MERRA2. That is, before 1998, the land surface in the model receives the precipitation of the following day.

Theoretically, this could enhance precipitation by increasing surface moisture supply before the precipitation actually falls. However, we have not been able to find a significant change before and after 1998. The error has been reported to NOAA's CPC.



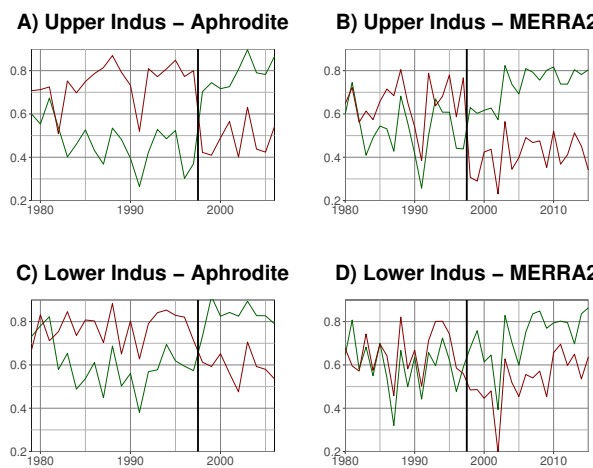

**Figure 3.** Daily correlation, per year, between CPC and Aphrodite (A and C), and MERRA2 (B and D) for both upper Indus (A and B) and lower Indus (C and D). The green line is the correlation between the same days in each datasets. For the red line, the previous day of CPC is used instead. The black vertical line is the start of the year 1998, around where the main error should be.





The comparison between TMPA and the rain gauge-based datasets APHRODITE and CPC (after 1998) shows a maximum of correlation delayed by around +3h, while no delays are evident for GPCC-daily and APHRODITE-2. This behaviour suggests that both CPC and APHRODITE are more representative of an accumulation period starting at 03hUTC, influenced by the Indian rain gauge network. APHRODITE-2 successfully corrected this delay, for a start at 00hUTC, like GPCC-daily. We also

found that most reanalyses have a negligible (<3h) delay with GPCC-daily. However, the reanalyses of the twentieth century have a different behaviour: both have a +12h delay with TMPA. For those two, only surface observations are assimilated. It is possible that 12h is the time needed by the troposphere to adjust to those surface constraints. Finally, we decided to take the accumulation period starting at 00h for all sub-daily datasets. Indeed, it is not straightforward to correct the delay in APHRODITE or CPC for instance, since only a daily resolution is available. Moreover, the correlation coefficients are not

significantly affected by those sub-daily lags.

We now start the comparison of the daily variability between each observational datasets for the upper Indus, using the correlation coefficients (upper part of Table 6). Note that the significance of the correlation (or of their difference) mentioned hereafter is defined for a 95% level. Two groups can be identified with higher correlation between the members of each. In the first group, TMPA and GPCP-1DD have a correlation of almost 0.9, showing how close those two datasets are, likely due to the

satellite observations they have in common and the similarity of retrieval procedures (Rahman et al., 2009; Palazzi et al., 2013; Rana et al., 2017). In the second group, the rain gauge-based datasets APHRODITE, CPC, and GPCC-daily have a correlation of around 0.8 between one another. The two versions of APHRODITE are even closer, due to their similarities of conception. When comparing GPCP-1DD and TMPA's correlation coefficients against the rain gauge-based datasets, it turns out that the TMPA coefficients are systematically significantly higher than those for GPCP-1DD. That is, TMPA variability is closer to the

rain gauge-based datasets than GPCP-1DD is. It could be either because TMPA includes more information from the rain gauge measurements than GPCP-1DD or because it has better quality (better algorithm, better raw data). Similarly, if we compare the rain gauge-based dataset coefficients against satellites, we note that APHRODITE has significantly higher values than CPC and GPCC-daily, as does APHRODITE-2.

We can argue that APHRODITE and APHRODITE-2 both represent the daily variability in the upper Indus better than CPC

and GPCC-daily using the correlation between the reanalyses and the observations (lower part of Table 6). Precipitation from reanalysis and observational data are independent from each other, in the sense that they do not share the same input data (except ERA5, which assimilates precipitation observations, and MERRA2, which integrate CPC data; the two need to be treated separately). Therefore, the correlations between the two types of datasets can only be explained by a common dependency on the true variability, which helps identifying the best datasets in each group. Moreover, those correlations have the same order

of magnitude as the correlations between each observational dataset. That is, reanalyses are as good as observational datasets in representing the daily variability.

For each reanalysis dataset, it is possible to rank the observational datasets depending on the correlation with the reanalysis (along the rows of Table 6). We found that APHRODITE-2 systematically ranks first, not far from APHRODITE, but with a correlation each time significantly higher than GPCC-daily, in third position. By contrast, CPC has systematically a lower rank

than GPCC-daily. We attribute the lower performance of CPC and GPCC-daily to the much lower number of observational





inputs than in APHRODITE and APHRODITE-2 (Table 2). Despite a slightly higher number of measurements, CPC performs worse than GPCC-daily, likely due to issues on the quality of those measurements, discussed in Section 3.2. Regarding satellite-based datasets, TMPA systematically outperforms GPCP-1DD, but the two, along with CPC, rank the lowest. That is, satellite measurements seem to perform poorly over the upper Indus.

We can also rank the correlation of the reanalysis products for each observational dataset (along the columns of table 6). ERA5 systematically ranks first. However, this reanalysis assimilates observations, such that it is not completely independent from the observational datasets. It is certainly a sign of good quality that the reanalysis output resembles the observations, but the reanalysis data could also include some of the observation errors, which is an issue that cannot be quantified by this analysis of the correlation. ERA-Interim ranks second, and is the best of the reanalysis not integrating observations. It

is closely followed by MERRA2, while CFSR has the lowest correlation with the observations of the latest generation of reanalyses. Interestingly for NCEP's reanalyses, the first outperforms the second version. The century reanalyses also show interesting behaviour: while 20CR has the lowest correlations with the observations, ERA20C performance is between CFSR and NCEP1, despite only assimilating surface observations. This behaviour clearly shows the progress made in reanalysis processing (e.g. in atmospheric modelling and data assimilation) over the last decades.

For the lower Indus domain, the results are quite similar (Table 7), but we also note some interesting differences. The correlations between the observations are all higher for this domain. In this flat domain, precipitation is less heterogeneous, and observations are more representative of their surrounding (i.e. larger spatial representativeness). In contrast, the reanalyses have lower correlations with observations than for the upper Indus. The lower Indus only receives precipitation during the summer monsoon, which is less well represented in models than the winter precipitation in the upper Indus (see below on

seasonality). More in details, APHRODITE-2 and APHRODITE still rank first, but the four other datasets rank in different orders: satellite products are possibly better in that flatter domain. For the reanalyses, we noticed that MERRA2 does not outperform MERRA1. Summer monsoon precipitation, especially in flat areas like this one, is strongly affected by surface moisture content (Douville et al. 2001). However, this parameter was not improved from MERRA1 to MERRA2 in the area as its authors would have expect by correcting the precipitation seen by the land surface model with CPC (Reichle et al., 2017,

Figure 1 in). On the contrary, we have shown that CPC likely underestimates precipitation (Table 4) and is not very good at representing the daily variability either (Table 7). This use of CPC possibly explains the slightly worse capability of MERRA2 compared to MERRA1, where improvement would have been expected. ERA5 and ERA-Interim remains the two reanalysis datasets with the highest correlation with the observations.



**Table 6.** Daily correlation between different datasets, in the upper Indus for the period 1998-2007.

| Datasets | APHRODITE | APHRODITE-2 | CPC | GPCC-daily | TMPA | GPCP-1DD |
|---|---|---|---|---|---|---|
| APHRODITE-2 | 0.92 | | | | | |
| CPC | 0.797 | 0.775 | | | | |
| GPCC-daily | 0.819 | 0.836 | 0.816 | | | |
| TMPA | 0.76 | 0.762 | 0.687 | 0.712 | | |
| GPCP-1DD | 0.735 | 0.725 | 0.665 | 0.676 | 0.898 | |
| ERA5 | 0.888 | 0.903 | 0.743 | 0.81 | 0.741 | 0.727 |
| ERA-Interim | 0.854 | 0.87 | 0.722 | 0.777 | 0.733 | 0.727 |
| JRA | 0.843 | 0.86 | 0.677 | 0.759 | 0.702 | 0.697 |
| MERRA2 | 0.846 | 0.862 | 0.714 | 0.778 | 0.708 | 0.699 |
| MERRA1 | 0.834 | 0.849 | 0.683 | 0.76 | 0.698 | 0.688 |
| CFSR | 0.795 | 0.82 | 0.64 | 0.74 | 0.641 | 0.625 |
| NCEP2 | 0.706 | 0.731 | 0.552 | 0.661 | 0.577 | 0.545 |
| NCEP1 | 0.76 | 0.769 | 0.606 | 0.687 | 0.619 | 0.598 |
| 20CR | 0.596 | 0.635 | 0.512 | 0.567 | 0.481 | 0.478 |
| ERA20C | 0.754 | 0.746 | 0.646 | 0.691 | 0.644 | 0.643 |





**Table 7.** Same as Table 6 for the lower Indus

| Datasets | APHRODITE | APHRODITE-2 | CPC | GPCC-daily | TMPA | GPCP-1DD |
|---|---|---|---|---|---|---|
| APHRODITE-2 | 0.887 | | | | | |
| CPC | 0.838 | 0.825 | | | | |
| GPCC-daily | 0.864 | 0.841 | 0.87 | | | |
| TMPA | 0.829 | 0.869 | 0.79 | 0.809 | | |
| GPCP-1DD | 0.771 | 0.801 | 0.72 | 0.74 | 0.906 | |
| ERA5 | 0.858 | 0.871 | 0.805 | 0.826 | 0.835 | 0.772 |
| ERA-Interim | 0.828 | 0.837 | 0.763 | 0.794 | 0.79 | 0.744 |
| JRA | 0.719 | 0.76 | 0.709 | 0.708 | 0.76 | 0.73 |
| MERRA2 | 0.777 | 0.794 | 0.723 | 0.763 | 0.725 | 0.677 |
| MERRA1 | 0.782 | 0.796 | 0.749 | 0.76 | 0.775 | 0.741 |
| CFSR | 0.7 | 0.69 | 0.626 | 0.657 | 0.672 | 0.618 |
| NCEP2 | 0.601 | 0.632 | 0.572 | 0.618 | 0.576 | 0.523 |
| NCEP1 | 0.635 | 0.643 | 0.605 | 0.623 | 0.596 | 0.545 |
| 20CR | 0.442 | 0.4 | 0.35 | 0.393 | 0.345 | 0.308 |
| ERA20C | 0.655 | 0.712 | 0.643 | 0.663 | 0.678 | 0.673 |





We investigated the possible seasonality in the correlation coefficients detailed so far for the upper Indus. We did not find notable variation between the observational datasets. Seasonal variation does occur when comparing the reanalyses to the observations. In Figure 4, we compare the daily variability of each reanalysis with APHRODITE-2 for each month. The data shows that the reanalyses are altogether more similar to APHRODITE-2 during the winter season than during summer, while

5   the differences between the reanalyses vary. From December to April, all reanalysis products have a similarly high correlation with the observational dataset (>0.9), except for the two century reanalyses, and to a lesser extent the older NCEP reanalyses. From May onwards, all correlations drop to various degrees. Both NCEP reanalyses drop the most, followed by CFSR. ERA5 shows the highest correlations, just above ERA-Interim, JRA, MERRA 1, and MERRA 2. For the century reanalyses, 20CR drops to very low values (<0.5 and even <0.2 in September and October), while ERA-20C remains at acceptable levels, around

10   CFSR. We therefore have very high confidence in the capability of most reanalyses to represent the daily variability in winter. In summer, the confidence is more dependent on the reanalysis, and overall lower than in winter. However, it is unclear if the seasonality of the correlation between the reference and the best reanalyses (ERA5, ERA-Interim, JRA, MERRA1, MERRA2) is due to a changing ability of the reanalyses or of the reference dataset, APHRODITE-2. The seasonality for those reanalyses disappears when using TMPA as a reference, but mainly due to a drop in winter correlation, which rather suggests that satellite

15   observations are not suited for that season (not shown). The analysis of the seasonality is less interesting in the lower Indus domain, since it is mainly dominated by the monsoon. The results resemble what was just discussed for summer in the upper Indus.





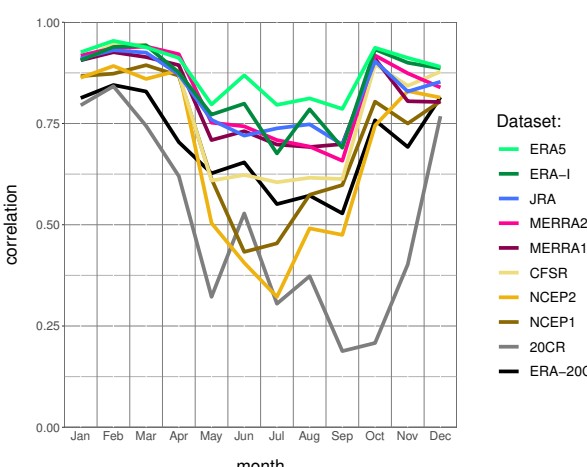

**Figure 4.** Daily correlation, per month, between APHRODITE-2 and each reanalysis, in the upper Indus. The period considered is 1998-2007.



We also looked at possible trends in the representation of the daily variability, due to a change in the type, quantity, or quality of input data in each dataset. We computed the time series of correlations between observations and reanalyses using a two year moving window for the upper Indus (Figure 5). However, the Pearson correlation we used so far is also known to be sensitive to extreme values. This leads to jumps in the correlation when an extreme value passes in the moving window and

is well predicted. In order to have a clearer signal, without jumps, we used instead the Spearman correlation. This coefficient is based on the rank rather than on the absolute value of each observation and is therefore not sensitive to extreme values. We checked that most of the results presented above are valid with the Spearman correlation as well.

In Figure 5-A, we compare the observational dataset using ERA-Interim as a reference. We first notice that APHRODITE and APHRODITE-2 always have significantly higher correlation scores than the others, except around 2004-2006, and relatively

stable values between 0.85 and 0.9. The quality of those two datasets found over the period 1998-2007 can therefore be extended to the whole period 1979-present. GPCC-daily exhibits stronger variability during the first 20-years, but then its score increases and stabilises around 0.85. This behaviour is likely due to an increase of the number of observations that are between 5 and 10 before 2000, but above 15 after 2005. CPC is in general very close to GPCC-daily, except around the year 2000, which explains the differences between the two datasets over the period 1998-2007 previously investigated. The two satellite products TMPA

and GPCP-1DD are very similar to each other, relatively stable, but at a lower level than the rain gauge-based datasets.

We now investigate the quality of the most recent reanalysis (Figure 5-B) using as reference APHRODITE (plain line) and APHRODITE-2 (dotted line). This reference is justified by the stability of the good result discussed above. The two references give similar results over their common period, which helps when analysing the whole time period. ERA5 and ERA-Interim are the two most stable reanalyses and have the highest correlations. JRA is also one of the best reanalysis datasets in the first

decade, but its correlation drops by about 0.05 compared to ERA5 after 1990 and never recovers. MERRA1 and 2 exhibit similar variability to each other, but the first version often has better results than the latter. CFSR is the most problematic reanalysis with the strongest variability and much lower correlation. However, it shows much better results at the end of the time period, with the release of its second version.

Finally, over the second half of the twentieth century, the large change in number and type of observations assimilated could

impact the quality of the reanalysis (Figure 5-C). However, no trend can be found. Correlations between JRA and APHRODITE remains mostly between 0.8 and 0.85. ERA-20C is also quite consistent over time, generally above NCEP1. 20CR, by contrast, exhibits a much higher variability with correlation dropping as low as 0.4 at times, and sometimes reaching NCEP1.

There are some differences in the results for the lower Indus (Figure 6). First, for the observation, CPC and GPCC-daily reach the quality of APHRODITE-2 around 2005, despite including half the number of observations (Figure 6-A). Certainly,

after 2005, the more homogeneous coverage of observations in CPC and GPCC-daily than in APHRODITE-2 counterbalances the reduction in number (Figure 1-D and E). Before 2005, the cause of the improvement of GPCC-daily can again be tracked to the increase in observations included, while the rise in quality of CPC remains of uncertain origin, since the number and location of observations are constant. TMPA shows correlation very close to CPC, with a similar unexplained rise between 2000 and 2005, almost reaching the quality of the rain gauge-based datasets. GPCP-1DD has lower scores than TMPA, but

also sees a rising trend during the two decades it covers. Comparing the differences between the reanalyses (Figure 6-B), we



found much smaller differences than when using the Pearson correlation (Table 7), which suggest that the difference in quality resides in the representation of the extreme events. No clear change can be observed during the period 1979-2015, however.

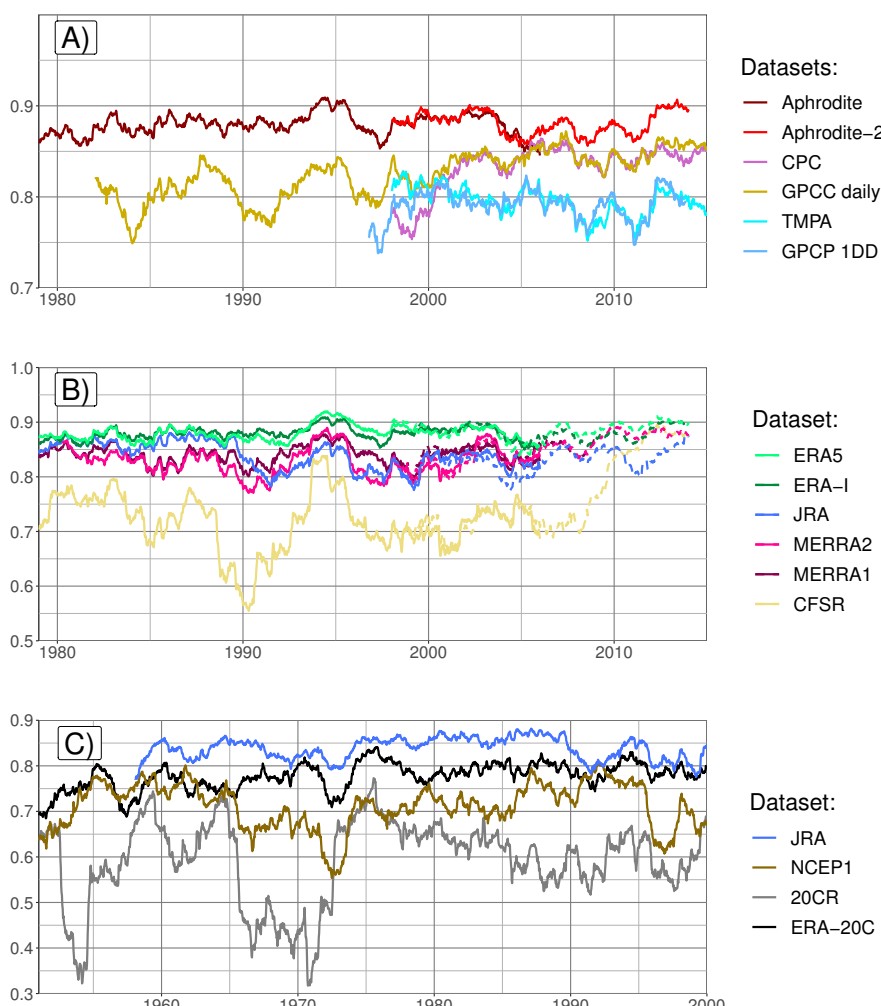

**Figure 5.** Daily correlation, on a running two-year window, between a reference and different datasets, for the upper Indus. The years on the x-axis is the start of the two-year window. In A) observational datasets are tested against ERA-Interim. Figure B) shows the correlation between a selection of reanalysis and APHRODITE over the period 1979-2007 (plain line) and APHRODITE-2 over the period 1998-2013 (dotted line). Finally, C) presents the reanalyses covering the second half of the 20th century, with APHRODITE as reference



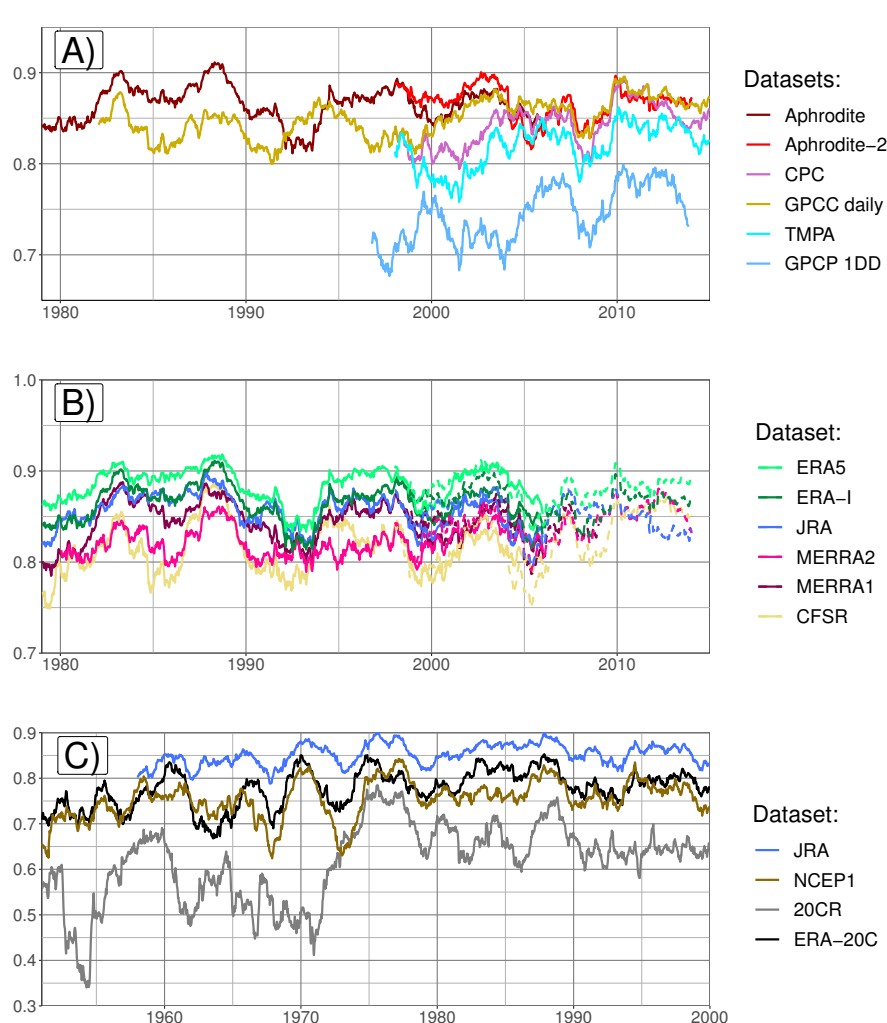

**Figure 6.** Same as Figure 5 but for the lower Indus



### 3.3 Monthly, seasonal, and inter-annual variability

A good representation of daily precipitation variability does not ensure a good representation of monthly or longer period variability. Moreover, all the observational datasets selected for this study can be analysed at a monthly time scale, such as GPCC-monthly or CRU, which only have a monthly resolution. In Figure 7, we present the trend in monthly correlation

between a reference and each type of dataset for the upper Indus. The correlation is calculated with the Pearson formula and over a ten-year moving window. It uses the monthly anomaly of precipitation, relative to a monthly mean computed over the same ten-year moving window. The reference to validate the observational datasets is ERA-Interim (A), and to validate the reanalyses GPCC-monthly (B). Those two datasets present lower variability in their quality and good results. They also cover the whole period 1979 to the present. However, we checked the main results with other references to validate them.

The best observational dataset for representing monthly variability for the upper Indus is APHRODITE (Figure 7-A). By contrast to the daily variability analysis, APHRODITE-2 has a much lower correlation with ERA-Interim on the common period with APHRODITE (1998-2007) and continues to drop after it. The difference in correlation between the two datasets is quite dependent on the reference, but all show the subsequent decrease. By contrast, CPC starts with the lowest correlation, but rises in the last decade at the level of the other datasets. CMAP, based on CPC also presents lower correlation, but is more

variable, and it depicts a similar rise around the year 2000. All the other datasets are very close to each other. CRU is slightly below GPCC-monthly, while GPCP products and TMPA, all including GPCC data, are slightly above it.

Still for monthly variability, the closest reanalysis to the observations is ERA5 (Figure 7-B), except when using CPC and CMAP as reference: then, MERRA2 has higher correlation at times, likely due the use of CPC data in both CMAP and MERRA2. The feedback impact of fixing the precipitation input in the surface model with CPC seems much more important

at this time scale than with the daily variability. Several datasets show a decrease in correlation during the 1990s: JRA, has a drop more pronounced than what is observed for the daily variability, and a drop appears for NCEP1, NCEP2 and ERA-20C. 20CR has the lowest correlation, while MERRA2, MERRA1, and ERA-Interim are quite similar, with correlation just below ERA5. CFSR also has relatively high values, but exhibits a decreasing trend, especially in the last 10 years, which is even more pronounced when testing with the other observational datasets. It is possible that version 2 of CFSR gives better results, but it

has not been running long enough to evaluate the monthly variability over a 10-year period. Instead, the correlations in Figure 7-B include both versions toward the end of the time period, which could add discrepancies when computing the monthly mean.

We also tested the datasets with the longest time coverage against GPCC-monthly (Figure 7-C). We found a relative consistency in the correlation with APHRODITE and CRU during the twentieth century: the timeseries do not diverge, despite

the lowering number of observations. However, since the datasets are not independent, we cannot say that the quality of those datasets remains constant. The reanalyses present fluctuating correlation with the reference. ERA-20C has lower correlation in the first half of the century, which could be due to a lowering confidence in either the reference or the reanalysis. However, ERA-20C correlation gets closer to 20-CR during that period, which suggests the variation in the reanalysis quality is the most important factor.





The lower Indus shows somewhat different results in terms of monthly variability (Figure 8). For the observations, APHRODITE does not have the highest correlations, as it is bypassed by GPCP-SG during the 1980s. After 2000, all datasets perform very similarly with two exceptions: CRU, which always has lower correlations, and APHRODITE-2 whose correlations drop during the last two years. CPC exhibits the same rising trend as for the upper Indus, but it is closer to the other datasets. For the re-

5    analysis, ERA5 still has the highest correlation but is joined by ERA-Interim just before the year 2000. MERRA2 also exhibits a rising trend. Surprisingly, MERRA2 does not show specifically higher correlation with CPC, as it does for the upper Indus domain, except for the two first years, where CPC has the lowest values. It is possible that the smaller difference in quality between CPC and the other observational datasets is not important enough to influence MERRA2's quality significantly. A drop in correlation is still observe for JRA, but it occurs latter, between 1995 and 2005. CFSR does not show any trend. NCEP1 has

10   surprisingly high correlations, especially during the 1980s when it is above CFSR. Finally, for the century-long datasets, CRU and GPCC-monthly show a decreasing trend, that could be related to an increasing difference in the observations included in each dataset. By contrast ERA-20C correlation drops at 20CR level before 1950.

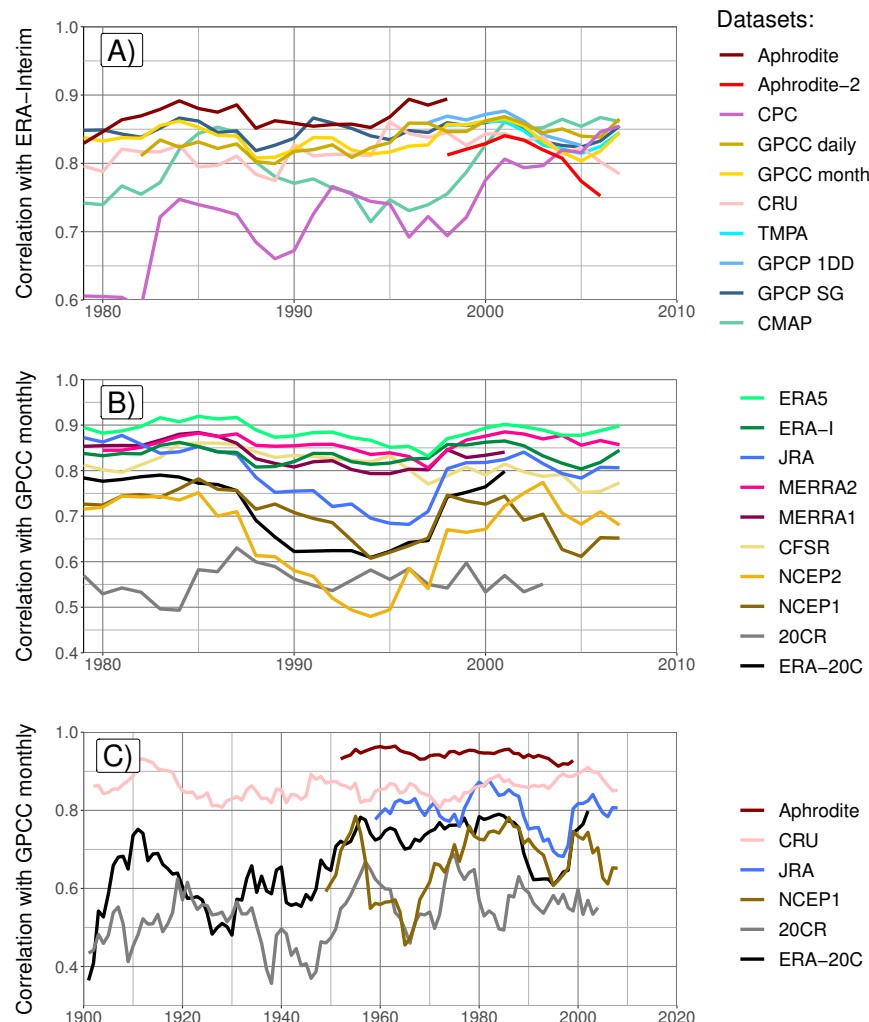

**Figure 7.** Correlation of monthly anomaly on a running ten-year window for the upper Indus. The monthly mean needed for the anomaly is computed relatively to the ten-year window. The years on the x-axis is the start of the ten-year window. Similarly as in Figure 5, a set of datasets is tested against a reference. In A) observational datasets are tested against ERA-Interim. B) shows the correlation between the reanalysis and GPCC-monthly. Finally, C) presents the longest datasets, except GPCC-monthly which is used as reference.



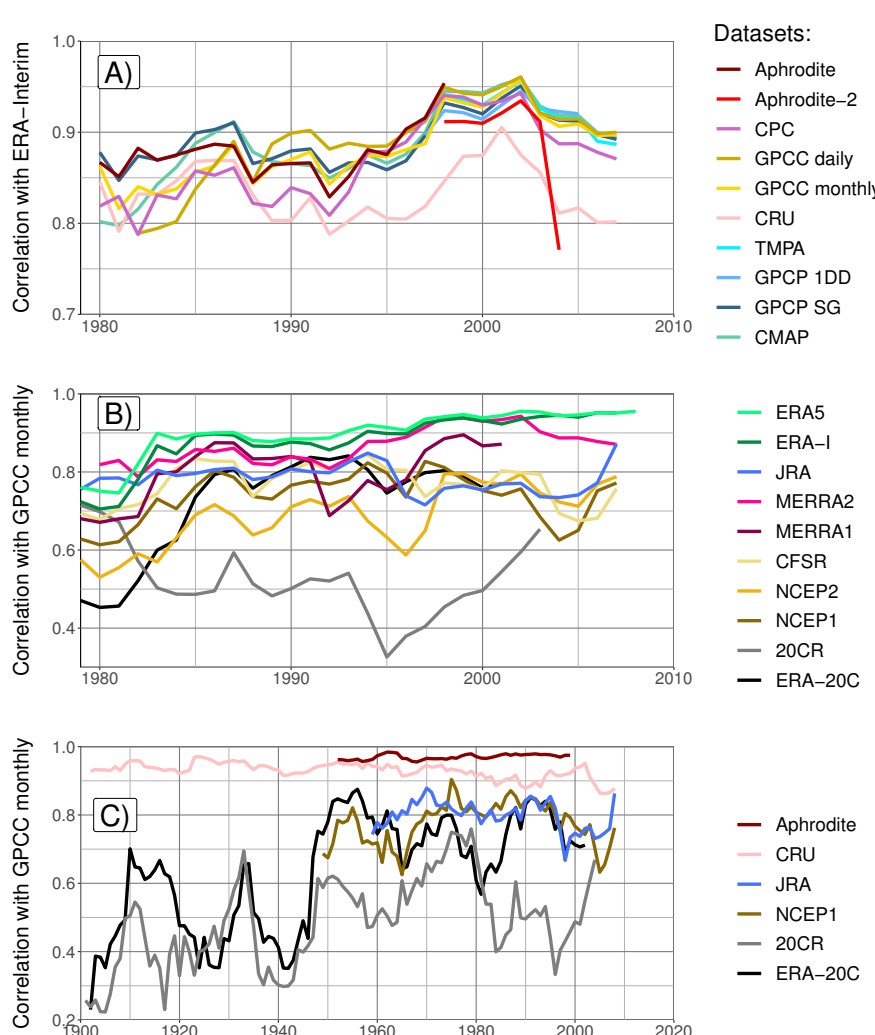

**Figure 8.** Same as Figure 7 but for the lower Indus





In Figure 9, we compare the inter-annual variability of GPCC-monthly to the reanalyses over the period 1981-2010 and to the other observational datasets covering that period. In Figure 10, we look at the 10-years moving mean for each of these datasets. The results are split by season and domain. GPCP inter-annual variability is almost identical to that of GPCC-monthly, due to the inclusion of GPCC-monthly data. By contrast, CPC has a much lower correlation with GPCC-monthly, especially in the upper Indus. This agrees with the lower capabilities found for the daily and monthly variability of CPC. Moreover, CPC is the most dissimilar observational dataset for the decadal variability, and is similar to CMAP and APHRODITE-2. In contrast, the other datasets show a very similar behaviour.

The reanalyses in winter have a behaviour similar to the observation for the period 1980-2010. Moreover, the most recent reanalyses tend to converge towards the same amount of precipitation after 2000. By contrast, the reanalyses that run before 1980 do not represent the decadal variability depicted by the observations. For summer in both domains, only ERA-5 represents the decadal variability observed. It also has an inter-annual correlation with GPCC-monthly that is higher than the correlation between GPCC and CRU. The other reanalyses have significantly lower correlation and miss all or some of the decadal variability. For example, in the upper Indus during summer, the precipitation amount increases after 2000 in the observations. While MERRA2 and CFSR show an increase of precipitation 2 or 3 times more important, ERA-interim and NCEP1 and 2 show instead a decrease. Moreover, only ERA5 and ERA-interim represent the peak around 1990 clearly. Similarly, in the lower Indus the 1990 peak is only reproduced by ERA5, CFSR, NCEP2, and ERA-20C, and the early twenty first century rise by ERA5 and ERA-Interim. Interestingly, while the observations show similar decadal variability for summer between the upper and the lower Indus, this is not the case for the reanalyses, except maybe for the twentieth century reanalyses, and of course ERA-5, which represent well the observations.





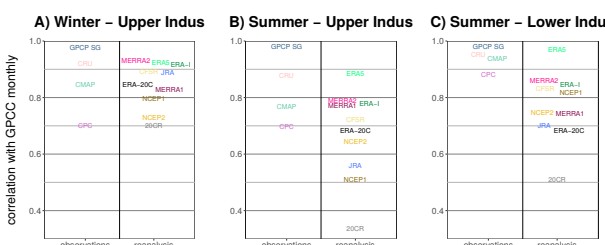

**Figure 9.** Inter-annual correlation on the period 1981-2010 between GPCC-monthly and the other datasets covering that period. The correlations are computed for specific seasons and domains. We split the result by type of dataset (Observation and renalysis)



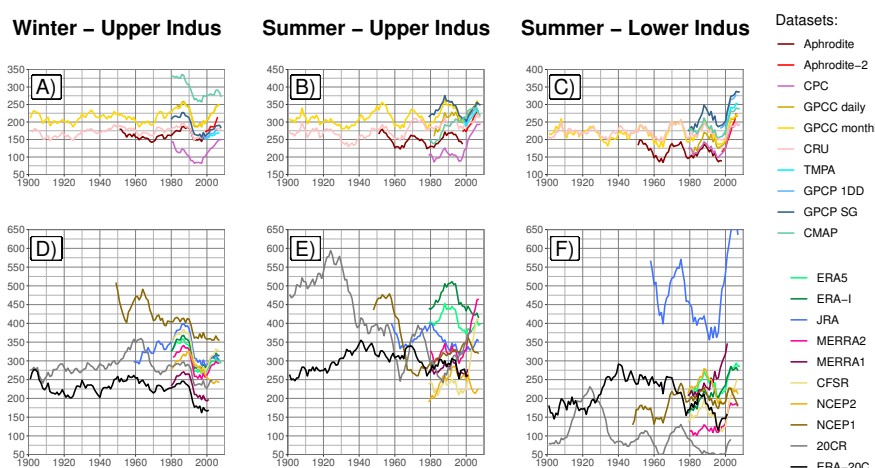

**Figure 10.** Decadal variability of precipitation using a 10-year running mean for different seasons and domains (Winter in the upper Indus: A and D; summer in the upper Indus: B and E; summer in the lower Indus: C and F) and the different datasets (Observational datasets: A, B and C; Renalysis datasets: D, E and F)



## 4 Conclusions

In this study, we have compared a large number of precipitation datasets of different types across two distinct zones of the Indus watershed: six based only on rain gauges, four derived from satellite observations, and ten from reanalysis. We have showed that the number and diversity of datasets help to identify and quantify the limitations and abilities of each, which in
turn enables a better estimation of the true values.

It is quite usual to validate reanalysis data using observations as a reference. When using daily correlation over a domain much larger than the spatial resolution, we found that most reanalyses are as similar to the observations as observational datasets are to each other. That is, reanalyses are as good as observational datasets for representing the daily variability, thereby justifying the possibility of using reanalyses as references for validating observations. The datasets validated can be
ranked depending on the correlation with the reference, and, if those datasets are independent from the reference, as are most reanalyses compared to observations, the ranking can be interpreted in terms of quality and proximity to the true variability. We found that the ranking is generally not affected by the reference, which increases the confidence in the results. Interestingly, the similarity between observations and reanalyses tends to decrease at lower frequencies, which we interpret as a difficulty of the reanalysis in representing the impact of larger scale drivers of precipitation (i.e. teleconnections). However, some reanalyses
remain good enough to be used as references at the monthly and inter-annual timescale.

The season also impacts the quality of datasets. All datasets represent the variability at all timescales very similarly during winter in the upper Indus. Only the satellite-based datasets represent the winter daily variability with more difficulty. In summer, however, in either of the domains, the correlations drop, primarily due to less reliable reanalyses, but also possibly less reliable rain gauge-based datasets. The dissimilarity is especially evident for the decadal variability, where only ERA5 represent the
same behaviour as the observations.

The method applied for the variability cannot be used to quantify the actual precipitation amount. Despite similar representation of the variability, the datasets exhibit important differences in the precipitation amount. We have relied on the literature to evaluate the different sources of uncertainty. Winter is an interesting season in the upper Indus, as the rain gauge-based datasets are known to underestimate snowfall significantly, especially in areas that are difficult to access. The reanalyses are not affected
by those biases, and, in light of their very good quality on the variability, we consider the precipitation amount in the reanalyses more realistic than in the observations. By contrast, precipitation amount during summer is much more uncertain, echoing the results on the variability. Notably, in some reanalyses the seasonality is delayed.

More specifically, for the observations, APHRODITE is the best dataset for daily and monthly variability, thanks to a large number of observations in the whole basin. However, it also exhibits drier conditions than most of the other datasets, which is
partially caused by the interpolation method it uses and possibly by a lower quality of the data. Surprisingly, APHRODITE-2 is not as good, especially for the longer term variability, as it removes some observations in areas with an already lower density of measurements. CPC is also a dry dataset, although this bias is reduced towards the end of the period covered. Similarly, the quality is much lower than other observational datasets during the 1980s and 1990s, and includes an error on the dates, but improves significantly after 2000. Since the number and location of measurements does not change, we suspect a change in the





quality of those measurements. GPCC-daily also sees an increase of daily correlation after 2000, and reaches APHRODITE-2 levels in the lower Indus. There, the improvement can be related to an increase in the number of measurements. Although it uses a very low number of measurements, its monthly mean is constrained by GPCC-monthly, which proves to be a good approach. Indeed, GPCC-monthly is one of the most reliable datasets in term of variability, but also in terms of amount as it is the only rain gauge-based dataset to include a correction for the measurement under-catchment, although the correcting factor is probably underestimated.

Satellite-based datasets are very dependent on the quality of the rain-gauge product they integrate. The added-value of satellite observations remains limited at the basin scale, and is probably more important in the flatter lower Indus, the summer season, and for longer term variability. Significantly, they do not correct the winter bias and show lower quality than rain-gauge based datasets on the daily correlation.

The reanalysis that represents best the observations is ERA5. It represents the variability at all timescales and the seasonality as well. It is the newest reanalysis and the only one to assimilate precipitation measurements. For that latter reason, however, it is somewhat problematic to use it to validate the observational datasets, although it does not give different results than other references. After ERA5, ERA-Interim, MERRA1 and MERRA2 have relatively similar performance. We found some dependency between MERRA2 and CPC, due to the use of CPC to correct the precipitation input on the land surface. This correction unlikely improved MERRA2's precipitation in the domain of study. JRA is relatively good over the period it covers, but exhibits a decrease in quality around 1990 in the upper Indus and 1995 in the lower Indus that lasts for around 10 years in each case. The summer seasonality is not very well reproduced either, especially in the lower Indus where JRA represents overly wet conditions. CFSR has problems reproducing the daily variability and the seasonality of the monsoon, especially in the upper Indus. This is probably improved by the latest version that started in April 2011. However, it would likely be better to treat the two versions separately as it seems the new version produces somewhat different statistics of precipitation. The twentieth century reanalyses, including only surface observation, are not as good as the others, especially in winter. However, while 20CR barely reproduces any of the variability depicted by the observation, ERA-20C has much better capabilities, close to NCEP1 and CFSR, especially during summer. Neither 20CR nor ERA-20C represent the decadal variability as shown by the observation before 1980.

This study has focused on the analysis of precipitation using basic tools such as mean and correlation. More complex tools also exist for a more thorough analysis of the precipitation statistics. Particularly, correlations are greatly impacted by extreme values, while important biases could also occur from inaccurate representations of the lowest precipitation rates. Moreover, we deliberately selected a large domain of study to improve the confidence in the datasets. Observation-based datasets still miss important patterns due to a lack of measurements in key areas, while reanalyses are even worse. However, either datasets could be used to constrain finer scale analysis.



*Code availability.* The code in R used to produce the figures and the tables is available upon request to the first author

*Author contributions.* Original idea, analysis and text by JPB, guidance and review by MH and CP

*Competing interests.* The authors declare that they have no conflict of interest

5 *Acknowledgements.* This research was carried out as part of the *TwoRains* project, which is supported by funding from the European Research Council (ERC) under the European Union's Horizon 2020 research and innovation programme (grant agreement no. 648609). We also thank our editor and reviewers for comments that improved the paper.





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
