# Peer review of "Cross-validating precipitation datasets in the Indus River basin"

_Hydrology and Earth System Sciences, 2019_

## Referee Comment (RC1) · Anonymous Referee #1 · 31 Jul 2019

Review of the paper: "Cross-validating precipitation datasets in the Indus River basin", by Jean-Philippe Baudouin, Michael Herzog, and Cameron A. Petrie

This paper presents a comprehensive assessment of several (twenty) state-of-the-art datasets for precipitation in the Upper Indus Basin region. These datasets cover different sources including station-based observational data, satellite products and re-analyses. The paper provides important and useful information on the strengths and uncertainties of the different datasets which could serve as a reference for precipitation in this area and be used for validation purposes.

However, in order to be really of use, the paper requires some revisions. Some sentences are hard to be read and should be rephrased or better explained. Some grammatical errors need to be fixed. I think that the paper deserves publication in this journal

only after the comments/requests listed below are carefully addressed.

*General comments/questions

- Is it possible from the analysis presented in the paper to really identify the best and/or the worst performing dataset, though a possible dataset "rank" probably depends on the variable/process one is looking at? The authors present advantages, disadvantages and strengths of the various datasets but the main message which remains to the reader, I think, is that many uncertainties remain. The conclusion, as it is, is not really "positive". What is the main message that the authors want to convey?

- Given the analysis presented in the paper, do the authors think that using the average of all datasets (a multi-dataset mean) could be a further output to be provided, along with the individual datasets themselves? I suggest to add the mean of all observation-based datasets, and of all reanalysis dataset indeed.

- One of the findings of the paper, which corroborates previous studies, is the fact that precipitation estimates from rain gauges are underestimated. The last sentence of the abstract highlights the need to account for this bias. Is it possible/reasonable, based on the study and results presented in the paper, to suggest "correction factors" to be applied to rain gauge estimates for the study area?

- In the description of the domain of study (section 2.1), it would be nice to have information on the average elevation of the two study domains and the range of elevations at least in the upper Indus.

- I have some concerns about the methodology used to interpolate the different datasets at the same spatial resolution. Is bilinear interpolation correct when dealing with precipitation fields? Wouldn't be preferable to use a more conservative approach? Did the authors test other approaches?

- In the section "Methods", the authors say that the comparison among the different datasets is performed in terms of mean and variability. What do they mean with "variability"? Is that year-to-year variability? Daily? Something else? This must be better specified in the Methods before going to the Results section.

- The discussion of Figure 2 (for the observation-based datasets, both in-situ and satellite), though containing many elements and considerations is, in my opinion, slightly confused. One reason is that the authors do not state in a very clear way that, e.g., they are taking one dataset as the reference (GPCC-monthly) against which to compare the other datasets. I agree that one reference is used, but this should be clearly stated (for example already in the "Methods" section)

- Each subsection of the "Results" section is very long (especially 3.1 and 3.2) and there is a risk that the reader gets "lost", in combination to the fact that there is a lot of information delivered. I suggest to try reducing these subsections a little bit and make it clear what the final message to the reader is. For example, one confusing thing, at least for me, is that on the one hand the "reference" dataset against which the other products are compared is GPCC-monthly (if I understood well), while, on the other hand, another cited paper (Dahri2018) is taken as a reference (long discussion in Section 3.1).

- Tables and Figures are not (always) correctly introduced in the text. In my opinion, when a figure/table is cited, it should be briefly described to say what it shows/displays (leaving the technical details to the caption and legend). Also check all Figures and Tables captions.

- Words like "consistency" or "consistent" are often used but I think that they are too generic. Please try to find other ways to convey the message.

- I'm not completely comfortable with the message that the reanalyses are useful to validate observations (as stated for example in the Conclusions). To be better discussed.

*Specific Comments

Abstract

- Page 1, Lines 3-4: In my opinion, the sentence "While rain gauge ... underestimation" would need to be rephrased. I would add a comma (,) after the word "reference" and I would change the subsequent sentence as follows, ", they provide information for specific, often sparse, locations (point observations) and are subject to underestimation in mountain areas", rather than "they are only punctual [...]"

- Page 1, Line 5: Add "data" after "reanalysis"

- Page 1, Line 10: Please replace "most able" with "most performing", or similar

- Page 1, Line 14: I don't understand whether "small" refers to "correction"; also, the term "correction" should be better explained. Please try to rephrase this sentence, if possible.

Introduction

- Page 2, Line 13: Add "of" between "use" and "rain gauges"

- Page 3, Line 1: 900km2 –> 900 km2

- Page 3, Line 2: 250 km2 –> 250 km2

- Page 3, Line 3: 15,000 km2 –> 15,000 km2

- Page 3, Line 7: 500 km2 –> 500 km2

- Page 3, Line 14: Add "the" between "and" and "heterogeneity"

- Page 3, Line 17: Remove "a" between "over" and "flat"

- Page 3, Line 18: Replace "those" at the end of the sentence with "station data"

- Page 3, Line 21: Replace "case" with "cases"

- Page 3, Line 23: Replace "consider" with "highlight" ; "reanalysis" with "reanalyses" and "observation" with "observations"

- Page 3, Line 25: Please change "has it made possible" with "has made it possible"

- Page 3, Line 26: Add "product" after "reanalysis"

- Page 3, Lines 28-30: I would remove the sentence starting with "Specifically" and ending with "variability". These are like results and conclusions of the study which is going to be presented, not useful here.

- Page 3, Line 30: Please replace "qualities" with e.g. "strengths and limitations"

- Page 3, Line 31: Please replace "have" with "has"

- Page 3, Line 34: Please replace "method" with "methods"

- Page 4, I would specify somewhere that the analyses described in items i), ii), and iii) concern precipitation. For example, "[...] which review the precipitation i) seasonal cycle [...], ii) daily variability [...], and iii) monthly and longer term [...]". Moreover, I would expect another sentence at the end of the paragraph for the Conclusions section. As it is, the sentence seems like suspended.

Data and methods

Domain of study

- Page 5. Line 8: I'm not sure to correctly identify the contour indicating the Luni River, as mentioned in the text. Is it the dark blue thick contour which also indicates the upper border of the study area? If so, as I also understand from the sentence at lines 10-11, I would better specify this at this point (maybe moving the sentence at lines 10-11 above)

- Page 5, Line 9: Please replace "bound" with "bounded"

- Page 5, Line 14: Please change "while the rest of the year it remains dry" with "while during the rest of the year the basin remains dry"

- Page 5, Line 16: I would replace "but exhibit" with "exhibiting"

- Page 5, Line 17: Rather than "process", I would say "circulation patterns"; also

please add another reference in parentheses which is significant for explaining the wintertime precipitation, i.e. Filippi et al., 2014 (Filippi, L., E. Palazzi, J. von Hardenberg, and A. Provenzale: Multidecadal Variations in the Relationship between the NAO and Winter Precipitation in the Hindu Kush-Karakoram. J. Climate, 27, 7890-7902, https://doi.org/10.1175/JCLI-D-14-00286.1, 2014)

- Page 5, Line 18: Please remove "it does"; the sentence is okay also in this way "As in the southern part of the basin"

- Page 5, Line 24: Not true, from Fig. 2, that there is no precipitation at all in winter in the lower Indus. I would rather say that "the southern part [...] is mostly characterized by summer precipitation (wintertime precipitation is negligible)", or something similar.

Data

- I would change the title of Section 2.2 in "Datasets", Section 2.2.1 in "Rain gauge data", Section 2.2.2 in "Satellite data", Section 2.2.3 in "Reanalysis data"

- Page 5, Line 29: Typo, there is a double parenthesis after the citation Yatagai et al., 2012.

- Page 5, Line 32: Please replace "the fact that" with "because"

- Page 6, Line 1: Delete "with a period covered". The sentence should be ... by the same institute extending up to 2015"

- Page 6, Line 7: Please replace "to" after "coverage" with "as"

- Page 6, Line 15: Please replace "of the available" with "among the available"

- Page 6, Line 15: What does "variety of input" mean? Please be more specific.

- Page 6, Lines 16-17. I don't understand what the sentence about GPCP_1DD means. Why do the authors specify that this specific dataset is valid only for comparison? What does this mean?

[Figure]

- Page 6, Line 18: At the beginning of the sentence, please rephrase "All three datasets described above use GPCC ...."

Reanalysis

- Page 9, Line 2: To avoid repetitions, "reanalysis datasets" should be replaced with "reanalysis data"

- Page 9, Line 3: better to say "can vary" rather than "varies"

- Page 9, Lines 3-4: I would rephrase this sentence in this way "Table 3 shows the ensemble of the ten reanalysis datasets which we used in this study".

- Page 9, Lines 10-11. I would change a little bit the sentence, for example: "ERA5 currently starts in 1979 (see Table 3) but future releases are expected to extend back to 1950".

- Page 9, Line 12: please change "than the others" with "than the other products"

Methods

- Page 12, Line 6: Please add "of precipitation" after "seasonality"

- Page 12, Line 9: "issues on" –> "issues of"

- Page 12, Line 9: For the last sentence, I would rather say "Winter is not analysed in the lower Indus as it is an extremely dry season"

- Page 12, Line 10: Please add "precipitation" before "time series"

- Page 12, Lines 11-18. For me all this doesn't fit here. This is already a result or, better, a possible explanation of the reasons why the different datasets show different behaviours. This should be discussed in the Results section, or in a dedicated Discussions section (to be eventually added) or in the Conclusions.

- Page 12, Line 22: "as we will discuss...." –> "as discussed in the Results section"

Results

Subsection 3.1

- Page 13, Line 3: Add "precipitation" before "seasonal cycle"

- Page 13, Line 13: The sentence of GPCC needs to be rewritten, I suggest: "... we compare the datasets to GPCC-monthly data, taken as a reference for this analysis". I think that it is more correct to state that GPCC-monthly is considered here as the reference rather than saying that it provides "good precipitation estimates", unless the authors add some references in support of this statement.

- Page 13, Line 7: I would start the sentence in a different way: "Figure 2 overall shows that all different datasets are able to capture the seasonality of precipitation in the two areas, though with different magnitudes"

- Page 13, Lines 7-12. I don't like the description of Figs. 2 A) and C) made in this paragraph. I would avoid sentences like "are ranked in the same order"; I would try to describe the climatology of precip. as seen by the different datasets taking one of them as the reference (as the authors do, if I understand well). Basically, the figure needs to be better described, also highlighting the performances of the various datasets in summer and winter.

- Page 13, Line 11: Please replace "inferior to" with "less than"

- Page 13, Line 13: I would replace "of mean precipitation" with "of the precipitation annual cycle"

- Page 13, Line 15: Regridding can be source of uncertainty, depending also on the kind of interpolation which is applied. The bilinear interpolation could not be the most appropriate method for precipitation. So the term "carefully" is questionable in this sentence, in my opinion.

- Page 13, Line 16: I would say that GPCC-daily "uses" GPCC-monthly and not that "is

based"

- Page 13, Lines 16-17: GPCC-daily and GPCC-monthly are not so different, as expected. Though GPCC-daily uses less stations, it incorporates GPCC-monthly analysis which uses more stations. I expect that these 2 products are very similar.

- Page 13, line 25: CPC is "drier" or driest? Maybe driest is the correct term and this refers to the upper Indus only.

- Page 13, Line 26: "linear relation"? I would rather say "clear correlation"

- Page 13, Line 30: I would replace "creator" with "developers"

- Page 13, Lines 31-33: please rephrase the entire sentence. Here is (only) a suggestion: "In particular, APHRODITE underestimation of total precipitation (compared to GPCC products) might be related to the fact that it partly relies on GTS data, in which missing values could be treated as no precipitation values. The large dry bias seen in CPC data could be associated with the same issue, since CPC is entirely based on GTS." I still don't understand, however, why a missing value in CPC would be treated as no precipitation.

- Page 14, Line 2: "build" –> "building"

- Page 14, Lines 2-5: This sentence needs to be rewritten, it is not really understandable especially when referring to TMPA, and to correlations (what datasets ?); this is really not clear to me.

- Page 14, Line 8: Please change "Different" with "Several" or "Various"

- Page 14, line 11: The sentence "they are basin-wide more numerous [...] territory" should be rephrased and improved.

- Page 14, Line 13: Rather than "explains" I would say "could reasonably explain"

- Page 14, Line 24: Remove "somewhat"

- Page 14, Line 25: "wetter by a factor of two". With respect to what? GPCC-monthly? To the observations in general? The reference has to be always indicated in a comparative sentence like this one.

- Page 14, Line 27: The sentence "some discrepancies are evident in the seasonality" could be misleading here, since only at line 33 the authors really report on changes in the seasonality, i.e., monthly shifts in some precipitation characteristics.

- Page 15, Line 5 (the whole paragraph). Besides the dry bias of rain gauges (raingauges are known to underestimate solid precipitation), one further reason for the wet bias of the reanalysis products (again, compared to GPCC-monthly) could be related to the "model component" of the reanalyses themselves. Models, in fact, are also known to have a wet (and cold) bias in mountains and in the cold season particularly (e.g., Palazzi et al., 2015; Palazzi, E., von Hardenberg, J., Terzago, S. et al. Clim Dyn (2015) 45: 21. https://doi.org/10.1007/s00382-014-2341-z). This should be added somewhere in the text. Reanalyses are a combination of observations+model which means that they can inherit drawbacks and advantages of both of them.

- Page 15, Line 33: "maxima" –> "maximum precipitation values"

- Page 16, Lines 4-5: I don't understand the meaning of the sentence "those errors are [...] yearlong", in particular of the term "consistent"

- Page 16, Line 5: Please consider to change this part "low density observations" with "a low density of observations"

- Page 16, Line 18: "over estimations" should be "overestimations"; "overruled", maybe better "avoided"?

- Page 16, Line 19: I would avoid qualitative expressions like "high to very high", just leave the percent values reported subsequently

- Page 16, Lines 20-21: "The summer mean does not converge", please rephrase this sentence. Do the authors mean that the spread among the various product is large?

- Page 16, Lines 23-24: In my opinion, the sentence starting with "These latest" and ending with "study domain" would be suitable as a final statement of this section.

Subsection 3.2

- Page 20, Line 2: Please add "precipitation" between "daily" and "variability". Same line: it is not clear to me what the concept of "dependency between each dataset" means

- Page 20, Line 4: Please replace "most of the reanalyses" with "of most of the reanalyses"

- Page 20, Line 4-5: the sentence in parentheses is unclear.

- Page 20, Lines 9-10: Please replace "from APHRODITE" to "APHRODITE-2" with "in both APHRODITE products"

- Page 22, Lines 28-29: I would rephrase this sentence: "common dependency of the true variability", in particular I'm not really comfortable with the term "true". The correlation between the two types of datasets can be related to the fact that they represent the precipitation variability at this scale in the same way?

- Page 23, Line9: "analysis of the correlation" –> "correlation analysis". Same line: I would say "ERA-Interim ranks second and is the best performing reanalysis among those which do not assimilate precipitation observations"

- Page 23, Line 11: Please add "version" between "first" and "outperforms". Same line: "century reanalysis" –> "20th century reanalysis" or the correct term for this product.

Subsection 3.3

- Page 32, Lines 3-4: Delete the part of the sentence after "time scale", not useful.

- Page 32, Line 8: "good", should be justified.

- Page 32, Line 12: Please add "the correlation" before "continues" (subject missing

here). Same at Line 14 ("it rises" or "the correlation rises")

- Page 32, Line 19: Remove "feedback". This sentence should be rephrased since it is not easily readable.

Conclusions

- Page 39, Line 3: "six" –> "six datasets are"; "four" –> "four are"

- Page 39, Line 4: "of datasets" –> "of the datasets"; "each"–> "each of them"

- Page 39, Line 5: "true values", an expression that should be avoided. It is quite clear, also from the analysis presented in this paper, that it is not possible to define a ground truth for precipitation, at least in this area.

- Page 39, Line 14: is there any reference to be cited in support to the statement about teleconnections?

- Page 39, Line 16: I would express the concept the other way around. For example "The quality of the datasets also depends on the season which is analysed"

- Page 39, Line 32: "CPC is also a dry dataset", I would rather say the "CPC exhibits a dry bias compared to ...."

- Page 40, Line 2. Is the word "There" at the beginning of the sentence used to say "In this case" (i.e., in the lower Indus)? I prefer "In this case" than "There".

- Page 40, last sentence: I suggest to rephrase this sentence, especially avoiding expressions like "while reanalyses are even worse". There are other ways to say that uncertainties remain. I would point more toward the lesson learned in this paper, with a more, let's say, positive view. That sentence is really sharp.

---

## Referee Comment (RC2) · Anonymous Referee #2 · 12 Aug 2019

The manuscript entitled "Cross-validating precipitation datasets in the Indus River basin" compares a collection of twenty rain gauge, satellite and reanalysis precipitation data sets in the upper and lower Indus river basin using a cross-validation methodology. This paper is a valuable study for academics and practitioners who use precipitation data sets in the area. My recommendation is that the paper is published after revision to the comments and questions below.

1) Abstract Line 14. "These findings highlight the need for a systematic characterisation of the underestimation of rain gauge measurements" Whilst you raise this issue in the abstract it is not discussed at all in the conclusions, either comment on this in the conclusion or remove from the abstract.

2) P.g.5. You provide a brief description of the catchment, but I think this could be

improved by stating actual elevation values of the catchment alongside the size of the catchment and the two subcatchments.

3) P.g.5. You use Figure 1 (A) as the reference in the description of the catchment, but I think more value would be obtained by making a separate larger figure to discuss the catchment. I think that the map should include elevation as well.

4) Section 2.2. You provide a very good description and rationale for why you selected certain rain gauge and reanalysis data sets. However, for the satellite data sets the section is very short. Was alternative satellite products considered, and if so why were they not picked? What was the advantage of selecting the data sets you do choose to include?

5) Page 6. Line 8 "which is useful for comparison" what do you mean by this comment? Are you saying that due to the CRU having a similar resolution and time coverage it was useful to compare to just the GPCC-monthly or for the entire analysis?

6) Page 6. Line 16 "and the largest variety of input" what do you mean by this comment?

7) Page 6. Line 17 "is useful for comparison" why is this data set in particular useful for comparison?

8) Page 6. Line 18 "All three datasets use GPCC for calibration" Which three data sets? Why is this important? Does this have any further implications in the analysis since the GPCC is used as the comparative data set?

9) Page 12. You use bi-linear interpolation to estimate the grids, why? Where other methods considered?

10) Page 12. How were abnormally large rainfall events (outliers) considered when you calculated the mean? As this may have skewed the mean?

11) Section 3. Whilst the results section is very extensive and detailed, it also is very

difficult to read due to it not having many (only 3) subsections. I think to improve you should split each of the subsections into subsubsections with their own theme.

12) Section 3. You use the GPCC-monthly data as the base to compare against however this was never justified in the text. I think this should be at least mentioned in Section 2.3 (methods) section.

13) Section 3. Partway through you change to compare against a different data set Dahri2018, why? Again this should be added into the methods section.

14) Page 40. Line 26 "Particularly, correlations are greatly impacted by extreme values". Why was this not discussed earlier in the text?

15) Page 40. Line 27 "Moreover, we deliberately selected a large domain of study to improve the confidence in the datasets" Why was this not discussed earlier in the text?

---

## Author Comment (AC1) · 1 Oct 2019

**1ˢᵗ REVIEWER**

*This paper presents a comprehensive assessment of several (twenty) state-of-the-art datasets for precipitation in the Upper Indus Basin region. These datasets cover different sources including station-based observational data, satellite products and reanalyses. The paper provides important and useful information on the strengths and uncertainties of the different datasets which could serve as a reference for precipitation in this area and be used for validation purposes. However, in order to be really of use,*

[Figure]

*the paper requires some revisions. Some sentences are hard to be read and should be rephrased or better explained. Some grammatical errors need to be fixed. I think that the paper deserves publication in this journal only after the comments/requests listed below are carefully addressed.*

**General comments/questions:**

*1) Is it possible from the analysis presented in the paper to really identify the best and/or the worst performing dataset, though a possible dataset "rank" probably depends on the variable/process one is looking at? The authors present advantages, disadvantages and strengths of the various datasets but the main message which remains to the reader, I think, is that many uncertainties remain. The conclusion, as it is, is not really "positive" . What is the main message that the authors want to convey?*

**Reply: In this study, the analysis of the quality of each dataset is limited to the precipitation in the Indus basin. We do not make presumptions about the quality of the datasets in other areas, nor other variables for the reanalyses. We have removed the term "rank" , particularly when performing the cross-correlation analysis (Section 3.3.2) as the reader may understand it in an absolute sense. We instead specify which datasets perform best for which measures.**

**Nevertheless, we can infer that, for example, reanalyses that represent the precipitation in the Indus basin better also represent circulation patterns and the processes involved in the generation of precipitation better.**

**Lastly, we specifically reorganised the conclusion around the key messages we want to convey. These are as follows:**

- **The method we have used gives detailed information on the strengths and**

**limitations of each of the 20 datasets investigated.**

- **There are large uncertainties, especially if considering all datasets equally. However, by evaluating the strengths and limitations of each dataset, we have found that some stand out as being of much higher quality which eventually helps to reduce the uncertainty.**

- **Particularly, progress in reanalysis products is real, with one (ERA5) scoring as high as the observations for all measures. These reanalyses offer a different point of view than the observations, which is useful for estimating the uncertainty, and can even be used to some extent to validate observations.**

- **We also emphasise on the need to systematically adjust rain gauge measurements to account for precipitation undercatchment.**

**TEXT MODIFIED**

- **For changes in Section 3.3.2 on cross-validation of Daily variability, which was part of Section 3.2 Daily variability in the reviewed document, see answer to general comment 8 of the first reviewer**

- **Changes in the Conclusion are shown below in bold:**

[revised manuscript text omitted]

*2) Given the analysis presented in the paper, do the authors think that using the average of all datasets (a multi-dataset mean) could be a further output to be provided, along with the individual datasets themselves? I suggest to add the mean of all observation-based datasets, and of all reanalysis dataset indeed.*

**Answer: analysing a multi-dataset mean does not provide more information about the quality of the datasets considered, and is therefore slightly outside the scope of the study. Such a mean is often used when uncertainties are very large and a best dataset cannot be selected (e.g. a multi-model mean for future simulation of the climate). In our study, however, we found that both APHRODITE and GPCC-monthly for the observation and ERA5 for the reanalyses were performing significantly better than other datasets. Furthermore, all datasets covered different periods, which complicates the use of a mean in practical situations.**

**Nevertheless, we checked if a mean can bring better results in terms of daily variability for the period 1998-2007. We considered the mean of the 6 observational datasets available at daily resolution, as well as the mean of ERA5, JRA, MERRA2 and CFSR (which are the most recent reanalyses). The correlation of these means against a reference are higher than that of most of the datasets composing them. However, the best datasets still have better scores. There is one exception in the lower Indus, as the mean of the observations performs significantly better than any of the individual observational datasets. In that domain, all observational datasets have very similar scores, and the mean is able to further improve these scores. These results seem to be too specific to be included in the study.**

*3) One of the findings of the paper, which corroborates previous studies, is the fact that precipitation estimates from rain gauges are underestimated. The last sentence of the abstract highlights the need to account for this bias. Is it possible/reasonable, based on the study and results presented in the paper, to suggest "correction factors" to be applied to rain gauge estimates for the study area?*

**Answer: This study does not evaluate the underestimation of rain gauge measurements, but rather suggests that this underestimation is the main cause of the differences between reanalyses and observations. Precipitation estimates from reanalyses are likely closer to the real amounts, but they may also be overestimated, and we cannot quantify this with our method. We rather urge a systematic correction of rain gauge measurements, using similar techniques as was used by Dahri et al. (2018), as this seems to be the best way to evaluate underestimations.**

**TEXT MODIFIED**

**See 6[th] paragraph of the updated conclusion in answer to general comment 1 of 1[st] reviewer**

*4) In the description of the domain of study (section 2.1), it would be nice to have information on the average elevation of the two study domains and the range of elevations at least in the upper Indus.*

**Answer: We added a new figure that gives the elevation, as suggested in comment 3 of the 2nd reviewer. It is used to introduce the domain of study.**

**TEXT MODIFIED:**

- **caption to figure 1 (See figure at the end):**

**"Relief and topographical features in and around the area investigated. The thick outer black contour represent the watershed on the Indus and Luni rivers. This area is split to form the two study areas: the upper Indus to the north, and the lower Indus to the south."**

- **At the start of section 2.1 "Study areas" , we have added (in bold):**

**"The Indus River basin extends across the north-westernmost part of the South Asian sub-continent, and is an area of various topographic features, as indicated in figure 1."**

*5) I have some concerns about the methodology used to interpolate the different datasets at the same spatial resolution. Is bilinear interpolation correct when dealing with precipitation fields? Wouldn't be preferable to use a more conservative approach? Did the authors test other approaches?*

**Answer: All rain-gauge datasets are based on a bi-linear interpolation of the station measurements, which justify the use of a bi-linear interpolation here. The only difference is that it is not the direct measurements that are interpolated in these observational datasets, but its anomaly against a climatology. However, using a specific climatology here would bias the validation.**

**Another, more conservative approach is to select the grid points whose centre is within the area of study. However, this would lead to small changes on the area being considered and precipitation biases. These biases are partly eliminated by the bi-linear interpolation.**

**TEXT MODIFIED**

  • **The whole of Section 2.3 Methods has been updated in light of other comments. The changes are in bold, the one considering this specific comment are in the first paragraph:**

"For each dataset, the time series of precipitation **are** averaged over the two **study areas (upper and lower Indus) and** calculated at a monthly resolution, and daily if possible. **The datasets have different spatial resolution which causes a problem when calculating the precipitation averages over the study areas. Simply selecting the cells whose centre is within these areas leads to small biases in the extent of the region considered. These biases are reduced by bi-linearly in-**

terpolating all data to a 0.25° grid, common to APHRODITE, APHRODITE-2, and GPCC-monthly. This choice is further discussed in section 3.1.1."

The analysis is performed over the 10-year period from 1998-2007, which is common to all datasets, **except when analysing the trends and inter-annual to decadal variability, for which we use all data available. We focus on the two wet seasons of the upper Indus.** Summer is defined from June to September, which matches the monsoon precipitation peak. Winter is defined from December to March. This fits the snowfall peak rather than the precipitation peak, but makes it possible to focus on issues of snowfall estimation (Palazzi et al. 2013). **In the lower Indus, we use the same definition of summer, but winter is not analysed, as it is a dry season.**

**We first compare the mean and seasonal cycle of each dataset in sections 3.1 and 3.2. There, for illustrative purposes, we make quantitative statements using GPCC-monthly as a reference. However, in section 3.1.3, we use the precipitation dataset from Dahri et al. (2018) as reference instead. This dataset cannot be used in other parts of the study, as it is limited to one part of the upper Indus, and only provides annual means.**

**Then, in section 3.3 we compare the daily variability of the precipitation using the Pearson correlation. The correlation significance is discussed at the 95% probability level. To reduce the impact of abnormally large rainfall events when investigating the trend (cf. Section 3.3.4), we use the Spearman correlation. Lastly, in section 3.4, other time scales of variability of the precipitation are investigated: monthly, seasonal, inter-annual, and decadal, still using the Pearson correlation at the 95% confidence interval."**

- **We also added a paragraph at the start of the Result section, on the differences of annual mean precipitation among rain gauges datasets (3.1.1), just after the introduction of the figures (cf. answer on comment 8 of the first reviewer)**

*6) In the section "Methods" , the authors say that the comparison among the different datasets is performed in terms of mean and variability. What do they mean with "variability" ? Is that year-to-year variability? Daily? Something else? This must be better specified in the Methods before going to the Results section.*

**Answer: the different time scales investigated are presented one paragraph above in that section, which led to some confusion for the readers. This has been corrected in the updated methodology (cf. Answer to the general comment 5 of the first reviewer).**

*7) The discussion of Figure 2 (for the observation-based datasets, both in-situ and satellite), though containing many elements and considerations is, in my opinion, slightly confused. One reason is that the authors do not state in a very clear way that, e.g., they are taking one dataset as the reference (GPCC-monthly) against which to compare the other datasets. I agree that one reference is used, but this should be clearly stated (for example already in the "Methods" section)*

**Answer: We added information about the use of references in the method section. We have also clarified in which part of the result sections which reference is used. Lastly, the discussion of figure 2 (now figure 3) on seasonality is performed differently and focuses only on seasonality and not on annual mean differences.**

**TEXT MODIFIED**

- **3rd paragraph of the reviewed Method section (cf. Answer to the general comment 5 of the first reviewer)**

- **The figure 3 on seasonality is discussed in a specific section (3.2, see answer to general comment 8 of the first reviewer)**

*8) Each subsection of the "Results" section is very long (especially 3.1 and 3.2) and there is a risk that the reader gets "lost" , in combination to the fact that there is a lot of information delivered. I suggest to try reducing these subsections a little bit and make it clear what the final message to the reader is. For example, one confusing thing, at least for me, is that on the one hand the "reference" dataset against which the other products are compared is GPCC-monthly (if I understood well), while, on the other hand, another cited paper (Dahri2018) is taken as a reference (long discussion in Section 3.1).*

**Answer: As suggested by the second reviewer, we split section 3.1 and 3.2 into several subsections (See comment 11 of the second reviewer).**

**The headings are now:**

**3.1 Annual mean**

**3.1.1 Differences between rain gauges-based datasets**

**3.1.2 Considering satellite and reanalysis datasets**

**3.1.3 Impact of rain gauge biases in mountainous terrains**

**3.2 Seasonal cycle**

**3.3 Daily variability**

**3.3.1 Lag analysis**

**3.3.2 Cross-validation**

**3.3.3 Influence of the seasonality**

**3.3.4 Trends**

**We particularly disentangled the analysis of the differences in annual mean on one hand, and the seasonal and monthly mean on the other. This improves the discussion of figure 3 on seasonality (cf general comment 7 and specific comment on Page 14, Line 27 of the first reviewer). We also put the comparison with Darhi et al. 2018 results in a specific subsection. Lastly, we deleted some redundant text.**

**TEXT MODIFIED**

- **We mention the use of Dahri et al. 2018 as a reference in the method section (cf. answer to general comment 5 of the first reviewer)**

- **New section 3.1 and 3.2 are now as follow, with the changes in bold:**

"

3. Results

**3.1 Annual mean**

[revised manuscript text omitted]

*9) Tables and Figures are not (always) correctly introduced in the text. In my opinion, when a figure/table is cited, it should be briefly described to say what it shows/displays(leaving the technical details to the caption and legend). Also check all Figures and Tables captions.*

**Answer: we added or changed the sentences that introduce each figures before discussing them, if this was not done properly**

**TEXT MODIFIED:**

- **Figure 1: "**The Indus River basin extends across the north-westernmost part of the South Asian sub-continent, **an area of various topographic features, as represented in figure 1."**

- **Figure 2: "Precipitation amount varies across the basin as shown in Figure 2-A."**

- **Table 1 and 2: "**We have selected five commonly used and one newly available gridded dataset based only on rain gauge data. **These are the** first six datasets **presented** in Table 1**. The mean number of stations used in the two study areas are available for five of the datasets and presented in Table 2."**

- **Table 3: "Table 3 shows the ensemble of the ten reanalysis datasets that have been used in this study."**

- **Table 4 and rest of Figure 2: "Annual mean of precipitation in both domains and for each dataset are given in Table 4 (last two columns). We first focus on the rain gauge-based datasets (upper part of the table). We found an important range of values in both study areas, which are related to dis-**

crepancies in the precipitation spatial pattern as presented in Figure 2-A to E."

- Figure 3: "The seasonal cycle of precipitation for each datasets is presented in figure 3."

- Table 6: "Table 6 presents the daily correlation of precipitation between the different datasets, for the upper Indus. The upper part of the table focuses on the cross-correlation between the observational datasets."

- Table 7: "The same correlation analysis is performed for the lower Indus domain (Table 7)."

- Figure 5: "Figure 5 presents the seasonality, for the upper Indus, of the correlations between the reanalyses and APHRODITE-2."

*10) Words like "consistency" or "consistent" are often used but I think that they are too generic. Please try to find other ways to convey the message.*

**Answer: We have found 3 occurrences of the words "consistent" and "consistency" . The first disappeared during the transformation of the first result section. In the other two cases, it had a meaning of stability.**

**TEXT MODIFIED**

- "Correlations between JRA and APHRODITE remains mostly between 0.8 and 0.85. ERA-20C is also **fairly stable** over time, generally above NCEP1. 20CR, by contrast, exhibits a much higher variability with correlation dropping as low as 0.4 at times, and sometimes reaching NCEP1"

- "We found **relatively stable correlations** with APHRODITE and CRU during the twentieth century"

*11) I'm not completely comfortable with the message that the reanalyses are useful to validate observations (as stated for example in the Conclusions). To be better discussed.*

**Answer: This is one of the key messages of the conclusion and is further discussed there (cf. answer to comment 1 of the 1ˢᵗ reviewer). In essence, both reanalysis and gridded observational data are estimates of the actual precipitation. Reanalyses rely the understanding of processes and a large ensemble of observations, while observational datasets only rely on precipitation measurements in specific locations. Their difference of approach makes them differently dependent on different sources of uncertainty, which allow a cross-validation. Importantly, the cross-validation is limited in our study to the precipitation variability at daily and monthly time scale and at the scale of the study areas. It is justified by the fact the most recent reanalyses represent variability within the range of uncertainty given by the different observational datasets. That is, these reanalyses represent the daily and monthly variability of the precipitation at least as well as the observational datasets.**

**Validation of the amount of precipitation is more complicated and needs a thorough understanding of the source of uncertainty in each dataset. Furthermore, we do not assume that the validation can be performed for other variables.**

**Specific Comments:**

**Abstract**

*Page 1, Lines 3-4: In my opinion, the sentence "While rain gauge ... underestimation" would need to be rephrased. I would add a comma (,) after the word "reference" and I would change the subsequent sentence as follows, ", they provide information for specific, often sparse, locations (point observations) and are subject to underestimation in mountain areas" , rather than "they are only punctual [...]"*

**Answer: Agreed, changed as suggested.**

*Page 1, Line 5: Add "data" after "reanalysis"*

**Answer: added**

*Page 1, Line 10: Please replace "most able" with "most performing" , or similar*

**Answer: we further clarified the sentence to "ERA5 is the reanalysis that offers estimates of precipitation closest to observations, "**

*Page 1, Line 14: I don't understand whether "small" refers to "correction" ; also, the term "correction" should be better explained. Please try to rephrase this sentence, if possible*

**Answer: "Correction" corresponds here to a factor by which the raw value are multiplied. "small" refers to that factor. We added the term "factor" twice in the sentence:**

"GPCC products are the only datasets that include a correction **factor** of the rain gauge
measurements but **this factor** remains likely too small"

**Introduction**

*Page 2, Line 13: Add "of" between "use" and "rain gauges"*
**Answer: added**

*Page 3, Line 1: 900km2 –> 900 km2*
*Page 3, Line 2: 250 km2 –> 250 km2*
*Page 3, Line 3: 15,000 km2 –> 15,000 km2*
*Page 3, Line 7: 500 km2 –> 500 km2*
**Answer: "2" is now a superscript.**

*Page 3, Line 14: Add "the" between "and" and "heterogeneity"*
**Answer: added**

*Page 3, Line 17: Remove "a" between "over" and "flat"*
**Answer: removed**

*Page 3, Line 18: Replace "those" at the end of the sentence with "station data"*
**Answer: replaced**

[Figure]

*Page 3, Line 21: Replace "case" with "cases"*

**Answer: replaced**

*Page 3, Line 23: Replace "consider" with "highlight" ; "reanalysis" with "reanalyses" and "observation" with "observations"*

**Answer: all replaced**

*Page 3, Line 25: Please change "has it made possible" with "has made it possible"*

**Answer: changed**

*Page 3, Line 26: Add "product" after "reanalysis"*

**Answer: added**

*Page 3, Lines 28-30: I would remove the sentence starting with "Specifically" and ending with "variability" . These are like results and conclusions of the study which is going to be presented, not useful here.*

**Answer: Agreed, deleted.**

*Page 3, Line 30: Please replace "qualities" with e.g. "strengths and limitations"*

**Answer: changed as suggested**

*Page 3, Line 31: Please replace "have" with "has"*

**Answer: replaced**

*Page 3, Line 34: Please replace "method" with "methods"*

**Answer: replaced**

*Page 4, I would specify somewhere that the analyses described in items i), ii), and iii) concern precipitation. For example, "[...] which review the precipitation i) seasonal cycle [...], ii) daily variability [...], and iii) monthly and longer term [...]" . Moreover, I would expect another sentence at the end of the paragraph for the Conclusions section. As it is, the sentence seems like suspended.*

**Answer: we have added a reference to the precipitation, as well as a sentence at the end of the paragraph in the conclusion section. We have updated the text to conform with the four (instead of three) subsections that now compose the Results section.**

**TEXT MODIFIED**

" The subsequent result section is split into **four** parts, which review, **for the precipitation: i) the annual mean, ii) the seasonality, iii) the** daily variability, and **iv) the** monthly and longer term variability. **The final section concludes with the main results, the potential of the method, and future research priorities.**"

**Data and methods**

**Domain of study**

*Page 5. Line 8: I'm not sure to correctly identify the contour indicating the Luni River,as*

*mentioned in the text. Is it the dark blue thick contour which also indicates the upper border of the study area? If so, as I also understand from the sentence at lines 10-11, I would better specify this at this point (maybe moving the sentence at lines 10-11above)*

**Answer: This is a misunderstanding, we do not provide the contour of the watershed of the Luni River. However, the Luni River is represented in Figure 1 (new figure, see answer to general comment 4 of the 1st reviewer), which illustrates our point.**

**The reference to the figure was misplaced and confusing. We have added a new sentence at the end of that paragraph to clarify the reference to the whole watershed.**

**TEXT MODIFIED**

"**It may also** forms seasonal rivers, such as the Luni River, which has been included in the **study areas**. This particular river flows into the Rann of Kutch, which is a flat salt marsh with complex connections with the Arabian Sea and the mouth of the Indus River (Syvitski et al., 2013), and is bounded on the west by the Aravalli Range. Although not strictly a part of the Indus watershed, it provides a clear and steady boundary for the domain of study. **The total watershed considered for the study is represented by the outer black line shown in figure 1**"

*Page 5, Line 9: Please replace "bound" with "bounded"*

**Answer: replaced**

*Page 5, Line 14: Please change "while the rest of the year it remains dry" with "while during the rest of the year the basin remains dry"*

**Answer: replaced**

*Page 5, Line 16: I would replace "but exhibit" with "exhibiting"*

**Answer: replaced as suggested**

*Page 5, Line 17: Rather than "process" , I would say "circulation patterns" ; also please add another reference in parentheses which is significant for explaining the wintertime precipitation, i.e. Filippi et al., 2014 (Filippi, L., E. Palazzi, J. von Hardenberg, and A. Provenzale: Multidecadal Variations in the Relationship between the NAO and Winter Precipitation in the Hindu Kush-Karakoram. J. Climate, 27, 7890-7902,https://doi.org/10.1175/JCLI-D-14-00286.1, 2014)*

**Answer: indeed, replaced as suggested. The reference has been added**

*Page 5, Line 18: Please remove "it does" ; the sentence is okay also in this way "As in the southern part of the basin"*

**Answer: good point, removed**

*Page 5, Line 24: Not true, from Fig. 2, that there is no precipitation at all in winter in the lower Indus. I would rather say that "the southern part [...] is mostly characterized by summer precipitation (wintertime precipitation is negligible)" , or something similar.*

**Answer: wrong phrasing indeed. We have modified the sentence.**

**TEXT MODIFIED**

"Thus, the northern part of the basin (hereafter the upper Indus, 595000 km$^2$) includes the maxima of precipitation along the Himalayas and most of the winter precipitation,

while the southern part (hereafter the lower Indus, 785000 km$^2$) is characterised **by a single wet season during summer, as wintertime precipitation is negligible**"

**Data**

*I would change the title of Section 2.2 in "Datasets" , Section 2.2.1 in "Rain gauge data" , Section 2.2.2 in "Satellite data" , Section 2.2.3 in "Reanalysis data"*

**Answer: Agreed, changed accordingly**

*Page 5, Line 29: Typo, there is a double parenthesis after the citation Yatagai et al.,2012.*

**Answer: thanks, removed**

*Page 5, Line 32: Please replace "the fact that" with "because"*

**Answer: replaced**

*Page 6, Line 1: Delete "with a period covered" . The sentence should be ... by the same institute extending up to 2015"*

**Answer: the sentence has been changed to:** " A new dataset has been issued in 2019 from the same institute **extending the period covered up to 2015**"

*Page 6, Line 7: Please replace "to" after "coverage" with "as"*

**Answer: replaced**

*Page 6, Line 15: Please replace "of the available" with "among the available"*

**Answer: replaced**

*Page 6, Line 15: What does "variety of input" mean? Please be more specific.*

**Answer: we referred to the amount and type of observations included in the datasets.**

**TEXT MODIFIED**

"It has the highest temporal and spatial resolution of the selection (sub-daily, and 0.25° like APHRODITE and GPCC-monthly) and **includes a large diversity of satellite observations**"

*Page 6, Lines 16-17. I don't understand what the sentence about GPCP_1DD means. Why do the authors specify that this specific dataset is valid only for comparison? What does this mean?*

**Answer: It does not make sense indeed, all datasets are compared, not particularly those ones. That part of the sentence is removed.**

**TEXT MODIFIED**

"The precipitation dataset from the Climate Research Unit has a similar resolution and time coverage as GPCC-monthly."

*Page 6, Line 18: At the beginning of the sentence, please rephrase "All three datasets described above use GPCC ...."*

**Answer: We specified the three datasets in the sentence.**

**TEXT MODIFIED**

"All three of these datasets **(TMPA, GPCP-1DD, and GPGP-SG)** use GPCC for calibration"

Reanalysis

*Page 9, Line 2: To avoid repetitions, "reanalysis datasets" should be replaced with "reanalysis data"*

**Answer: replaced**

*Page 9, Line 3: better to say "can vary" rather than "varies"*

**Answer: agreed, replaced**

*Page 9, Lines 3-4: I would rephrase this sentence in this way "Table 3 shows the ensemble of the ten reanalysis datasets which we used in this study" .*

**Answer: Replaced, this is part of the general comment 9 of the 1st reviewer**

*Page 9, Lines 10-11. I would change a little bit the sentence, for example: "ERA5 currently starts in 1979 (see Table 3) but future releases are expected to extend back to 1950" .*

**Answer: We updated the sentence.**

**TEXT MODIFIED**

**"ERA5 currently starts in 1979 but future releases are expected to extend this back to 1950."**

*Page 9, Line 12: please change "than the others" with "than the other products"*
**Answer: changed**

**Methods**

*Page 12, Line 6: Please add "of precipitation" after "seasonality"*
**Answer: Does not apply in the improved version of that section**

*Page 12, Line 9: "issues on" –> "issues of"*
**Answer: replaced**

*Page 12, Line 9: For the last sentence, I would rather say "Winter is not analysed in the lower Indus as it is an extremely dry season"*
**Answer: we did not add "extremely" here. Being dry is enough not to have analysed that season**

**TEXT MODIFIED**

**"In the lower Indus, we use the same definition for summer, but winter is not**

**analysed, as it is a dry season"**

*Page 12, Line 10: Please add "precipitation" before "time series"*

**Answer: Does not apply in the improved version of that section**

*Page 12, Lines 11-18. For me all this doesn't fit here. This is already a result or,better, a possible explanation of the reasons why the different datasets show different be-haviours. This should be discussed in the Results section, or in a dedicated Discus-sions section (to be eventually added) or in the Conclusions.*

**Answer: agreed, the passage is removed**

*Page 12, Line 22: "as we will discuss...." –> "as discussed in the Results section"*

**Answer: Does not apply in the improved version of that section**

Results

Subsection 3.1

*Page 13, Line 3: Add "precipitation" before "seasonal cycle"*

**Answer: Does not apply in the improved version of that section**

*Page 13, Line 3: The sentence of GPCC needs to be rewritten, I suggest: "... we*

*compare the datasets to GPCC-monthly data, taken as a reference for this analysis"
.I think that it is more correct to state that GPCC-monthly is considered here as the
reference rather than saying that it provides "good precipitation estimates" , unless the
authors add some references in support of this statement.*

**Answer: Agreed, we have now justified the use of qualitative statement using
GPCC as a reference for illustrative purposes. This is now explained in the meth-
ods section (cf. answer to general comment 5 of the 1ˢᵗ reviewer).**

**TEXT MODIFIED**

**In the method section: " We first compare the mean and seasonal cycle of each
datasets in sections 3.1 and 3.2. For quantitative statements we use GPCC-
monthly as a reference"**

*Page 13, Line 7: I would start the sentence in a different way: "Figure 2 overall shows
that all different datasets are able to capture the seasonality of precipitation in the two
areas, though with different magnitudes"*

**Answer: Does not apply in the improved version of that section**

*Page 13, Lines 7-12. I don't like the description of Figs. 2 A) and C) made in this
paragraph. I would avoid sentences like "are ranked in the same order" ; I would try to
describe the climatology of precip. as seen by the different datasets taking one of them
as the reference (as the authors do, if I understand well). Basically, the figure needs
to be better described, also highlighting the performances of the various datasets in
summer and winter.*
**Answer: To clarify the description, this is now presented in a dedicated section (3.2, cf. answer to general comment 8 of the 1st reviewer). We also focus on the months with a minimum or a maximum of precipitation, as well as on bias specific to a season, or more stable throughout the year.**

**TEXT MODIFIED**

**See answer to general comment 8 of the 1st reviewer**

*Page 13, Line 11: Please replace "inferior to" with "less than"*

**Answer: Does not apply in the improved version of that section**

*Page 13, Line 13: I would replace "of mean precipitation" with "of the precipitation annual cycle"*

**Answer: Does not apply in the improved version of that section**

*Page 13, Line 15: Regridding can be source of uncertainty, depending also on the kind of interpolation which is applied. The bilinear interpolation could not be the most appropriate method for precipitation. So the term "carefully" is questionable in this sentence, in my opinion.*

**Answer: This referred to the fact that we could have used the grid points whose centre was in the domain, instead of a bi-linear interpolation, which leads to further biases (cf. answer to general comment 5 of the 1st reviewer). But the word "carefully" does not reflect this idea, and is therefore removed. This point is further discussed in a specific paragraph (cf. general comment 5 of the 1st reviewer)**

**TEXT MODIFIED**

See second paragraph of section 3.1.1 in answer to general comment 8 of the 1st reviewer

*Page 13, Line 16: I would say that GPCC-daily "uses" GPCC-monthly and not that "is based"*

**Answer: This sentence is slightly modified in the improved version of the manuscript, and takes account of the comment.**

**TEXT MODIFIED**

**"The two GPCC products [...] uses the same climatology."**

*Page 13, Lines 16-17: GPCC-daily and GPCC-monthly are not so different, as expected. Though GPCC-daily uses less stations, it incorporates GPCC-monthly analysis which uses more stations. I expect that these 2 products are very similar.*

**Answer: Indeed, we have rephrased this point. We expect the datasets to be similar, but some differences exist, which should be related to the interpolation method we used, and this can be investigated.**

**TEXT MODIFIED**

See second paragraph of section 3.1.1 in answer to general comment 8 of the 1st reviewer

*Page 13, line 25: CPC is "drier" or driest? Maybe driest is the correct term and this refers to the upper Indus only.*

**Answer: CPC is the driest dataset for the upper Indus, and the second driest for the lower Indus. This is now explicitly stated in the text.**

**TEXT MODIFIED**

**"For example, CPC is by far the driest dataset in the upper Indus and the second driest in the lower Indus. This is likely related to the low number of observations it includes,** leaving vast areas with no or very few observations, including the wettest regions (Figure 2-E)"

*Page 13, Line 26: "linear relation" ? I would rather say "clear correlation"*

**Answer: Correlation refers to a specific statistical tool, which we did not use here. We change the word "relation" to "relationship"**

*Page 13, Line 30: I would replace "creator" with "developers"*

**Answer: changed**

*Page 13, Lines 31-33: please rephrase the entire sentence. Here is (only) a suggestion: "In particular, APHRODITE underestimation of total precipitation (compared to GPCC products) might be related to the fact that it partly relies on GTS data, in which missing values could be treated as no precipitation values. The large dry bias seen in CPC data could be associated with the same issue, since CPC is entirely based on GTS." I still don't understand, however, why a missing value in CPC would be treated as no precipitation.*

**Answer: There is a misuse of the word "treating" , we have replaced it with "misreporting" . The whole sentence was not very clear either. So, zero values can be reported by mistake, and the quality checks do not identify them as missing values. This is clarified in the text.**

**TEXT MODIFIED**

**"They noted that APHRODITE partly** relies on GTS data that are sent in near real time to the global network**, with risks of misreporting. The risk particularly concerns misreported zero values, harder to detect and which could lead to a dry bias. The large dry bias seen in CPC data could be associated with the same issue, since CPC is entirely based on GTS data"**

*Page 14, Line 2: "build" –> "building"*

**Answer: replaced**

*Page 14, Lines 2-5: This sentence needs to be rewritten, it is not really understandable especially when referring to TMPA, and to correlations (what datasets ?); this is really not clear to me.*

**Answer: This is a minor point and it is removed from the text.**

*Page 14, Line 8: Please change "Different" with "Several" or "Various"*

**Answer: We used the word "several" as suggested.**

*Page 14, line 11: The sentence "they are basin-wide more numerous [...] territory"*
*should be rephrased and improved.*

**Answer: The sentence is split into two**

**TEXT MODIFIED**

"However, the difference in mean precipitation is most likely related to the change in observations from rain gauges. **Although the APHRODITE-2 comprises more observations basin-wide**, this increase mainly happens over Indian territory."

*Page 14, Line 13: Rather than "explains" I would say "could reasonably explain"*

**Answer: changed accordingly**

*Page 14, Line 24: Remove "somewhat"*

**Answer: Does not apply in the improved version of that section**

*Page 14, Line 25: "wetter by a factor of two" . With respect to what? GPCC-monthly?To the observations in general? The reference has to be always indicated in a comparative sentence like this one.*

**Answer: We modified the sentence so we actually compare JRA with 20CR, making the point about the large spread of values.**

**TEXT MODIFIED**

**"the wettest dataset, JRA, is five times wetter than the driest dataset, 20CR"**

*Page 14, Line 27: The sentence "some discrepancies are evident in the seasonality" could be misleading here, since only at line 33 the authors really report on changes in the seasonality, i.e., monthly shifts in some precipitation characteristics.*

**Answer: This comment lead us to split the subsection "Seasonal cycles and annual means" in two, one addressing the annual means and biases, and the second the seasonal cycles and seasonal biases. See answer to comment 7 of the 2nd reviewer.**

*Page 15, Line 5 (the whole paragraph). Besides the dry bias of rain gauges (rain-gauges are known to underestimate solid precipitation), one further reason for the wet bias of the reanalysis products (again, compared to GPCC-monthly) could be related to the "model component" of the reanalyses themselves. Models, in fact, are also known to have a wet (and cold) bias in mountains and in the cold season particularly(e.g., Palazzi et al., 2015; Palazzi, E., von Hardenberg, J., Terzago, S. et al. Clim-Dyn (2015) 45: 21. https://doi.org/10.1007/s00382-014-2341-z). This should be added somewhere in the text. Reanalyses are a combination of observations+model which means that they can inherit drawbacks and advantages of both of them.*

**Answer: Agreed, and this is also suggested by the fact reanalysis overestimate the annual amount found in Dahri2018.**

**TEXT MODIFIED**

- **In section 3.1.3 (See answer to general comment 8 of the 1st reviewer)**

**"Nevertheless, the four selected reanalysis datasets in Table 5 overestimate the Dahri2018 adjusted value, by 20% on average. This suggests that part but not all of the differences between reanalysis and observational data can be explained**

by biases from the latter. Modelled precipitation in reanalysis are likely overesti-
mated in the upper Indus, which corroborates results from previous studies (e.g.
Palazzi et al., 2015). "

• **Further in that section:**

"Reanalyses tend to be wetter than observational datasets in the upper Indus,
which is partly explained by the underestimation of the observations."

*Page 15, Line 33: "maxima" –> "maximum precipitation values"*

**Answer: Does not apply in the improved version of that section**

*Page 16, Lines 4-5: I don't understand the meaning of the sentence "those errors
are[...] yearlong" , in particular of the term "consistent"*

**Answer: We meant that the errors are relatively independent of the season. It is
removed in the improved version of that section**

*Page 16, Line 5: Please consider to change this part "low density observations" with"a
low density of observations"*

**Answer: Does not apply in the improved version of that section**

*Page 16, Line 18: "over estimations" should be "overestimations" ; "overruled" , maybe
better "avoided" ?*

**Answer: Does not apply in the improved version of that section**

*Page 16, Line 19: I would avoid qualitative expressions like "high to very high" , just leave the percent values reported subsequently*

**Answer: Agreed, this is removed in the improved version of that section**

*Page 16, Lines 20-21: "The summer mean does not converge" , please rephrase this sentence. Do the authors mean that the spread among the various product is large?*

**Answer: Does not apply in the improved version of that section**

*Page 16, Lines 23-24: In my opinion, the sentence starting with "These latest" and ending with "study domain" would be suitable as a final statement of this section.*

**Answer: Agreed, the last sentence of section 3.1 is now an updated version of this one.**

**TEST MODIFIED**

**Last sentence of section 3.1:**

**"Lastly, all datasets suffer from spatial discrepancies, which are** detrimental to small-scale comparisons, especially near mountains, but justify our choice to use a larger study **area"**

**Subsection 3.2**

**Note that this corresponds to subsection 3.3 in the new update manuscript**

*Page 20, Line 2: Please add "precipitation" between "daily" and "variability" .*

**Answer: added**

*Same line: it is not clear to me what the concept of "dependency between each dataset" means*

**Answer: Our sentence is not related to the lag analysis that follows, it is moved to the new section 3.3.2 "Cross-validation in the upper Indus" . We replaced the word "dependency" with "co-variability" , which is investigated in that section using correlations.**

**TEXT MODIFIED**

   • **The section 3.3.1 starts by:**

**"Investigating the daily precipitation variability helps to better quantify the quality of each dataset."**

   • **The section 3.3.2 starts by:**

We now start the comparison of the daily variability **between each dataset. Particularly, we aim to understand whether the co-variability exhibited between datasets is** coming from the use of common methods or **data source**, or from a good representation of the **precipitation variability.**

*Page 20, Line 4: Please replace "most of the reanalyses" with "of most of the reanaly-*
*ses"*

**Answer: Does not apply in the improved version of that section**

*Page 20, Line 4-5: the sentence in parentheses is unclear.*

**Answer: TMPA has a sub-daily resolution and we can compute a 24-hour accumulation ending at different times of the day (e.g. 0h, 3h, 6h ... UTC). APHRODITE has daily-accumulated precipitation values, and we would expect the accumulation period to end at 00h UTC. To check this, we test whether APHRODITE values correlate better with TMPA values when these ones are accumulated up to 21h, 0h, or 3h, and so on. The problematic sentence is removed from the paragraph, and we have revised the paragraph discussing the sub-daily lag.**

**TEXT MODIFIED**

**In the paragraph discussing the sub-daily lag:**

**"Possible differences in the End of Day times of the observational datasets are investigated using the sub-daily resolution of TMPA. We compute TMPA daily accumulation with different End of Day time and determine which one maximises the correlation with the other observational datasets. APHRODITE and CPC (after 1998) maximise the correlation with TMPA when for the latter an End of Day at 03h UTC is used."**

*Page 20, Lines 9-10: Please replace "from APHRODITE" to "APHRODITE-2" with "in both APHRODITE products"*

**Answer: There is a misunderstanding here, the two APHRODITE datasets have actually an opposite characteristic (on the consideration of different End of Day**
time). We have rephrased the sentence.

**TEXT MODIFIED**

**"Neither GPCC-daily nor APHRODITE documentation mention this issue, while a specific effort has been made to homogenise all observations in APHRODITE-2."**

*Page 22, Lines 28-29: I would rephrase this sentence: "common dependency of the true variability" , in particular I'm not really comfortable with the term "true" . The correlation between the two types of datasets can be related to the fact that they represent the precipitation variability at this scale in the same way?*

**Answer: The correlation between the two types of datasets is without doubt related to the fact that they represent the precipitation variability at this scale in the same way, or rather, the correlation is a measure of that similarity. The question here is about the cause of that similarity: it is either because they are based on a similar method or input data, or solely because they both try to estimate precipitation. This is further explained at the start of section 3.3.2**

**TEXT MODIFIED**

- **1st paragraph of section 3.3.2:**

"We now start the comparison of the daily variability **between each dataset. Particularly, we aim to understand whether the co-variability exhibited between datasets is** coming from the use of a common method or **data source**, or from a good representation of the **precipitation variability. All datasets are estimates of precipitation, but they use different methods and input data to achieve this (cf. section 2.2). If**

two datasets share a similar method or data source, this could at least partly explain the co-variability between the datasets. If, on the contrary, the two datasets are independent, then the co-variability they share is most likely due to the precipitation signal they estimate. Therefore, the higher the correlation between two independent datasets, the better the estimate of precipitation in both datasets."

- The problematic sentence is modified:

"Therefore, the correlations between the two types of datasets **is not affected by common data or method, and is rather a measure of their quality,** which helps identifying the best datasets in each group**."**

*Page 23, Line9: "analysis of the correlation" –> "correlation analysis" .*

**Answer: This sentence is removed**

*Same line:I would say "ERA-Interim ranks second and is the best performing reanalysis among those which do not assimilate precipitation observations"*

**Answer: we further modified this sentence to avoid the use of the word "rank"**

**TEXT MODIFIED**

"ERA-Interim has the **second highest correlations**, and is the best performing reanalysis among those that do not assimilate precipitation observations"

*Page 23, Line 11: Please add "version" between "first" and "outperforms" .*

**Answer: added**

*Same line:"century reanalysis" –> "20th century reanalysis" or the correct term for this product.*

**Answer: we added the word "twentieth"**

Subsection 3.3

**Note that this corresponds to section 3.4 in the new version**

*Page 32, Lines 3-4: Delete the part of the sentence after "time scale" , not useful.*

**Answer: agreed, deleted**

*Page 32, Line 8: "good" , should be justified.*

**Answer: The good quality is demonstrated in that section**

**TEXT MODIFIED**

"Those two datasets present a more stable quality and good **correlations as we demonstrate below.**

*Page 32, Line 12: Please add "the correlation" before "continues" (subject missing here). Same at Line 14 ("it rises" or "the correlation rises" )*

**Answer: added**

*Page 32, Line 19: Remove "feedback" . This sentence should be rephrased since it is not easily readable.*

**Answer: The sentence has been removed**

Conclusions

*Page 39, Line 3: "six" –> "six datasets are" ; "four" –> "four are"*

**Answer: changed**

*Page 39, Line 4: "of datasets" –> "of the datasets" ; "each" –> "each of them"*

**Answer: changed**

*Page 39, Line 5: "true values" , an expression that should be avoided. It is quite clear,also from the analysis presented in this paper, that it is not possible to define a ground truth for precipitation, at least in this area.*

**Answer: We replaced "true value" with "uncertainty"**

*Page 39, Line 14: is there any reference to be cited in support to the statement about teleconnections?*

**Answer: Removed from the updated version of the conclusion. We want to highlight the fact that ERA5 does represent decadal variability.**

*Page 39, Line 16: I would express the concept the other way around. For example"The quality of the datasets also depends on the season which is analysed"*

**Answer: we rephrased this sentence.**

**TEXT MODIFIED**

**"We also found that the quality of the datasets depends on the season."**

*Page 39, Line 32: "CPC is also a dry dataset" , I would rather say the "CPC exhibits a dry bias compared to ...."*

**Answer: The new sentence is further improved, but we have considered the comments.**

**TEXT MODIFIED**

**"CPC [...] with a large dry bias compared to GPCC-monthly"**

*Page 40, Line 2. Is the word "There" at the beginning of the sentence used to say "In this case" (i.e., in the lower Indus)? I prefer "In this case" than "There" .*

**Answer: Does not apply in the improved version of that section**

*Page 40, last sentence: I suggest to rephrase this sentence, especially avoiding expressions like "while reanalyses are even worse" . There are other ways to say that uncertainties remain. I would point more toward the lesson learned in this paper, with a more, let's say, positive view. That sentence is really sharp.*

**Answer: We changed the last paragraph to emphasise several points: large uncertainty remains, but some datasets perform better, cross-validation between reanalysis and observational datasets are possible, and we suggest future possibilities such as quality monitoring.**

Please also note the supplement to this comment:
https://www.hydrol-earth-syst-sci-discuss.net/hess-2019-303/hess-2019-303-AC1-supplement.pdf
* * *
[Figure]

**Fig. 1.**

**Supplement:**

---

## Author Comment (AC2) · 1 Oct 2019

**2nd REVIEWER**

*The manuscript entitled "Cross-validating precipitation datasets in the Indus River basin" compares a collection of twenty rain gauge, satellite and reanalysis precipitation data sets in the upper and lower Indus river basin using a cross-validation methodology. This paper is a valuable study for academics and practitioners who use precipitation data sets in the area. My recommendation is that the paper is published after revision to the comments and questions below.*

[Figure]

*1) Abstract Line 14. "These findings highlight the need for a systematic characterisation of the underestimation of rain gauge measurements" Whilst you raise this issue in the abstract it is not discussed at all in the conclusions, either comment on this in the conclusion or remove from the abstract.*

**Answer: We have added a paragraph concerning this issue in the conclusion as it is one of the key messages we want to convey (cf. answer to general comment 1 of the 1st reviewer)**

**TEXT MODIFIED**

• **In the conclusion:**

**"As mentioned above, rain gauge-based datasets underestimate precipitation. Only GPCC products use a correction factor to account for measurement underestimation, but this one is still too small. We emphasise the need to correct directly the measured values before interpolation to a grid, using, for example, methods similar to those developed by Dahri et al. (2018)"**

*2) P.g.5. You provide a brief description of the catchment, but I think this could be improved by stating actual elevation values of the catchment alongside the size of the catchment and the two sub-catchments.*

**Answer: we added the size of the domain considered in that section. We also added a figure with the elevation (cf. answer to comment 4 of the 1st reviewer).**

**TEXT MODIFIED**

- **On the size of the domains (section 2.1):**

"Thus, the northern part of the basin (hereafter the upper Indus, **595000 km$^2$**) includes the maxima of precipitation along the Himalayas and most of the winter precipitation, while the southern part (hereafter the lower Indus, **785000 km$^2$**) is characterised **by a unique wet season, in summer, as wintertime precipitation is negligible"**

- **Regarding the elevation map, see answer to general comment 4 of the first reviewer.**

*3) P.g.5. You use Figure 1 (A) as the reference in the description of the catchment, but I think more value would be obtained by making a separate larger figure to discuss the catchment. I think that the map should include elevation as well.*

**Answer: The figure has been added and is used to introduce the study areas (cf answer to comment 4 of the 1$^{st}$ reviewer and 2 of 2$^{nd}$ reviewer)**

*4) Section 2.2. You provide a very good description and rationale for why you selected certain rain gauge and reanalysis data sets. However, for the satellite data sets the section is very short. Was alternative satellite products considered, and if so why were they not picked? What was the advantage of selecting the data sets you do choose to include?*

**Answer: The main reason we selected those datasets is that they enable the study over a common period of 10 years with the other dataset as they start in 1998 or earlier. We added a sentence at the start of that subsection to discuss this point.**

**TEXT MODIFIED**

- **At the start of section 2.2.2:**

**"Various satellite-based gridded precipitation products are available, but we have only selected datasets providing data from 1998, to ensure a long enough common period with the rain gauge-based datasets (the common period reaches years due to APHRODITE ending in 2007)."**

*5) Page 6. Line 8 "which is useful for comparison" what do you mean by this comment?Are you saying that due to the CRU having a similar resolution and time coverage it was useful to compare to just the GPCC-monthly or for the entire analysis?*

**Answer: this has been removed as the aim of the study is to compare all datasets, not specifically those (same specific comment by 1st reviewer)**

*6) Page 6. Line 16 "and the largest variety of input" what do you mean by this comment?*

**Answer: we have referred to the amount and type of observations included in the datasets (same specific comment by 1st reviewer).**

*7) Page 6. Line 17 "is useful for comparison" why is this data set in particular useful for comparison?*

**Answer: Similarly as for comment 5 of the 2nd reviewer, this sentence has been modified.**

[Figure]

**TEXT MODIFIED**

**"We also selected the daily product from the Global Precipitation Climatology Project (GPCP-1DD; Huffman and Bolvin, 2013) as well as the monthly product issued by the same group (GPGP-SG Adler et al., 2016)"**

*8) Page 6. Line 18 "All three datasets use GPCC for calibration" Which three datasets? Why is this important? Does this have any further implications in the analysis since the GPCC is used as the comparative data set?*

**Answer: We added the name of three datasets in parenthesis. It does impact the analysis when using GPCC as reference. This characteristic is used to explain the results where appropriate. We added a sentence here to clarify this point. We also changed the following sentence to highlight the reason for the selection of CMAP (unlike the other it is calibrated by another dataset, this further addresses the comment 4 of the 2nd reviewer).**

**TEXT MODIFIED**

- **At the end of section 2.2.2:**

**"**All three of these datasets **(TMPA, GPCP-1DD, and GPGP-SG)** use GPCC for calibration**, which could introduce some similarities. By contrast, the last dataset** included, CPC Merged Analysis of Precipitation (CMAP; Xie and Arkin, 1997), uses CPC for calibration**. It** has the same time coverage and resolution as GPCP-SG**. This version** does not include reanalysis data, to simplify the analysis.**"**
*9) Page 12. You use bi-linear interpolation to estimate the grids, why? Where other methods considered?*

**Answer: Same comment as general comment 5 of the 1st reviewer, see answer there.**

*10) Page 12. How were abnormally large rainfall events (outliers) considered when you calculated the mean? As this may have skewed the mean?*

**Answer: Abnormally large rainfall events are not treated separately. If limited to a small area, their effect is mitigated by taking the average over the study areas. We justify this as we are not interested in such fine scale phenomena in this study. Some abnormal events remain at the basin scale. They cause problems when considering correlation on a moving window (e.g. Figure 5 and 6). In that case we have used the Spearman coefficient. We also checked that the main result remained the same when the Pearson coefficient is used. A sentence is added to the method section about the use of Spearman correlation. We also further discussed the limitation of the Pearson correlation regarding extreme values in the conclusion**

**TEXT MODIFIED**

- **in the Method section:**

**"To reduce the impact of abnormally large rainfall events when investigating the trend (cf. Section 3.3.4), we use the Spearman correlation."**

- **in the conclusion**

"We have used the Pearson correlation to compare the datasets, although it has some limitations. For example, it is affected by extreme values, that is, in our context, abnormally large precipitation events. These led to some difficulties in interpreting trends and we preferred the Spearman formula in this context (cf. Figures 6 and 7). By contrast, the Pearson correlation is less affected by the difficulties in representing the lowest precipitation rates, although these one could explain some of the biases."

*11) Section 3. Whilst the results section is very extensive and detailed, it also is very difficult to read due to it not having many (only 3) subsections. I think to improve you should split each of the subsections into subsubsections with their own theme.*

**Answer: We agree with the comment and split the subsection 1 and 2 of the result section into several sub-subsection. See answer to the general comment 8 of the first reviewer**

*12) Section 3. You use the GPCC-monthly data as the base to compare against however this was never justified in the text. I think this should be at least mentioned in Section 2.3 (methods) section.*

**Answer: Similar to comment 7 and 8 of the first reviewer. We added information about the use of references in the method section**

*13) Section 3. Partway through you change to compare against a different data set- Dahri2018, why? Again this should be added into the methods section.*
**Answer: The Dahri2018 dataset offers a very interesting assessment of the rain gauge undercatchment and demonstrates that part of the difference in mean precipitation between reanalyses and observational datasets can be explained by this undercatchment. However, we cannot use this dataset as a reference in the whole study, as it covers only a small fraction of the Indus watershed, which is itself included in our upper Indus domain. Furthermore, the paper does not provide the monthly values that we could have used to assess the seasonality. The part of the analysis using this dataset is now enclosed in a subsection, and we refer to it in the method section. See also answer general comment 8 of the first reviewer, and the changes in the method section in the answer to general comment 5 of the first reviewer**

*14) Page 40. Line 26 "Particularly, correlations are greatly impacted by extreme values". Why was this not discussed earlier in the text?*

**Answer: This sentence has been removed. We actually found similar results using both Spearman and Pearson correlation coefficients. What is more problematic is the heavy tail towards 0, and we further discuss this in the conclusion (see answer to comment 10 of the 2nd reviewer)**

*15) Page 40. Line 27 "Moreover, we deliberately selected a large domain of study to improve the confidence in the datasets" Why was this not discussed earlier in the text?*

**Answer: we discuss this point now in the conclusion in relation to the important uncertainty in the fine scale spatial patterns of precipitation. See 4th paragraph of the updated conclusion in the answer to the general comment 1 of the first reviewer.**